# CXCR3-expressing myeloid cells recruited to the hypothalamus protect against diet-induced body mass gain and metabolic dysfunction

**Natalia Mendes**[1,2], **Ariane Zanesco**[2], **Cristhiane Aguiar**[3], **Gabriela F Rodrigues-Luiz**[4], **Dayana Silva**[2], **Jonathan Campos**[2], **Niels Olsen Saraiva Camara**[5], **Pedro Moraes-Vieira**[3], **Eliana Araujo**[2,6*†], **Licio A Velloso**[2,7*†]

[1]School of Medical Sciences, Department of Translational Medicine (Section of Pharmacology), University of Campinas, Campinas, Brazil; [2]Laboratory of Cell Signaling, Obesity and Comorbidities Research Center, University of Campinas, Campinas, Brazil; [3]Laboratory of Immunometabolism, Institute of Biology - University of Campinas, Brazil, Campinas, Brazil; [4]Department of Microbiology, Immunology and Parasitology, Federal University of Santa Catarina, Florianópolis, Brazil; [5]Laboratory for Transplantation Immunobiology, Institute of Biomedical Sciences, University of Sao Paulo, Sao Paulo, Brazil; [6]Faculty of Nursing, University of Campinas, Campinas, Brazil; [7]National Institute of Science and Technology on Neuroimmunomodulation, Rio de Janeiro, Brazil

**\*For correspondence:**
earaujo@unicamp.br (EA);
lavellos@unicamp.br (LAV)

[†]These authors contributed equally to this work

**Competing interest:** The authors declare that no competing interests exist.

## eLife Assessment

This work is of **fundamental** significance and has an **exceptional** level of evidence for a new population that protects against obesity-induced hypothalamic inflammation. This topic will attract attention from a broad base of readers, from hypothalamic neuroscientists to immunologists with an interest in metabolism.

**Abstract** Microgliosis plays a critical role in diet-induced hypothalamic inflammation. A few hours after a high-fat diet (HFD), hypothalamic microglia shift to an inflammatory phenotype, and prolonged fat consumption leads to the recruitment of bone marrow-derived cells to the hypothalamus. However, the transcriptional signatures and functions of these cells remain unclear. Using dual-reporter mice, this study reveals that CX3CR1-positive microglia exhibit minimal changes in response to a HFD, while significant transcriptional differences emerge between microglia and CCR2-positive recruited myeloid cells, particularly affecting chemotaxis. These recruited cells also show sex-specific transcriptional differences impacting neurodegeneration and thermogenesis. The chemokine receptor CXCR3 is emphasized for its role in chemotaxis, displaying notable differences between recruited cells and resident microglia, requiring further investigation. Central immunoneutralization of CXCL10, a ligand for CXCR3, resulted in increased body mass and decreased energy expenditure, especially in females. Systemic chemical inhibition of CXCR3 led to significant metabolic changes, including increased body mass, reduced energy expenditure, elevated blood leptin, glucose intolerance, and decreased insulin levels. This study elucidates the transcriptional differences between hypothalamic microglia and CCR2-positive recruited myeloid cells in diet-induced inflammation and identifies CXCR3-expressing recruited immune cells as protective in metabolic

outcomes linked to HFD consumption, establishing a new concept in obesity-related hypothalamic inflammation.

## Introduction

Obesity affects over 600 million people worldwide and projections are pessimistic indicating that over a billion people will be diagnosed with obesity by the year 2030 (https://www.worldobesity.org/). Obesity develops as a consequence of a chronic state of anabolism in which caloric intake overcomes energy expenditure (*Theilade et al., 2021*). Both, experimental and human studies have indicated that, at least in part, the chronic anabolic state leading to obesity develops as a consequence of a defective hypothalamic regulation of whole-body energy balance (*Cavadas et al., 2016*; *Sonnefeld et al., 2023*; *van de Sande-Lee et al., 2020*).

Experimental studies have shown that the consumption of large amounts of saturated fats triggers an inflammatory response in the hypothalamus affecting the function of key neurons involved in the regulation of food intake, energy expenditure, and systemic metabolism (*De Souza et al., 2005Milanski et al., 2009*; *Zhang et al., 2008*; *Milanski et al., 2012*). In addition, human studies using magnetic resonance imaging have identified hypothalamic gliosis in adults and children with obesity, providing clinical evidence for the existence of an obesity-associated hypothalamic inflammation (*van de Sande-Lee et al., 2020*; *Thaler et al., 2012*; *Sewaybricker et al., 2019*). Microglia are key cellular components of this inflammatory response undergoing structural and functional changes that develop early after the introduction of a high-fat diet (HFD) (*Tapia-González et al., 2011*; *Valdearcos et al., 2014*; *Fernández-Arjona et al., 2022*; *Valdearcos et al., 2017*). Distinct strategies used to inhibit hypothalamic microglia resulted in impaired diet-induced hypothalamic inflammation, increased leptin sensitivity, reduced spontaneous caloric intake, and improved systemic glucose tolerance (*Valdearcos et al., 2014*; *Morari et al., 2014*). Thus, elucidating the mechanisms involved in the regulation of microglial response to dietary factors may provide advance in the definition of the pathophysiology of obesity, and potentially identify new targets for interventions aimed at treating obesity and its metabolic comorbidities (*Mendes et al., 2018*; *Mendes and Velloso, 2022*; *Valdearcos et al., 2019*).

Microglia are derived from the yolk sac primitive hematopoietic cells and populate the neuroepithelium at embryonic day 9.5 (*Ginhoux et al., 2010*). Under baseline conditions, in the absence of infections, trauma or other types of potentially harmful stimuli, these cells remain rather quiescent; however, under stimulus, they undergo rapid morphological and functional changes aimed at confronting the threat (*Saijo and Glass, 2011*). In the hypothalamus, differently from most parts of the brain, microglia are responsive to fluctuations in the blood levels of nutrients and hormones involved in metabolic control, such as leptin (*Valdearcos et al., 2014*). Thus, a simple meal can promote considerable changes in hypothalamic microglia indicating the involvement of these cells in the complex network of cells that regulate whole-body metabolism (*Gao et al., 2014*). Under the consumption of a nutritionally balanced diet, meal-induced changes in hypothalamic microglia are cyclic and completely reversible (*Gao et al., 2014*). However, under consumption of a HFD, microglia present profound changes in morphology and function; moreover, upon prolonged consumption of this type of diet, there is recruitment of bone marrow-derived myeloid cells that will compose a new landscape of hypothalamic microglia (*Valdearcos et al., 2014*; *Morari et al., 2014*; *Valdearcos et al., 2019*). Despite the considerable advance in the understanding of how hypothalamic microglia are involved in diet-induced obesity (DIO), it is currently unknown what are the transcriptional landscapes of resident microglia and recruited myeloid cells and what specific functions they exert in the context of obesity and metabolic control.

In this study, we first elucidated the transcriptional differences between resident CX3CR1+ hypothalamic microglia and CCR2+ recruited myeloid cells in DIO, identifying chemokines as a relevant subset of genes undergoing regulation. Next, we identified a subset of recruited myeloid cells expressing CXCR3, which exerts a protective role against DIO.

## Results

### Elucidating the transcriptional signatures of hypothalamic resident microglia and recruited myeloid cells in DIO

Dual-reporter mice CX3CR1GFP/+CCR2RFP/+ were obtained by the crossing of CX3CR1GFP and CCR2RFP homozygous mice (*Figure 1a*). At the age of 8 weeks, female and male were randomly selected for either chow or HFD feeding for 28 days, and then specimens were harvested for analysis (*Figure 1b*). In flow cytometry, cells expressing CCR2 were detected in the hypothalamus of mice fed on HFD (*Figure 1c*), whereas in the white adipose tissue (WAT), virtually all cells expressing CX3CR1 were also expressing CCR2 (*Figure 1c*). Conversely, in both female and male mice fed on HFD, the ratio of CCR2 myeloid cells to CX3CR1 resident microglia was approximately 1:10 (*Figure 1d*). Histological analysis confirmed the results obtained by flow cytometry (*Figure 1e*); furthermore, it was shown that CD169, which is classically regarded as a marker of bone marrow-derived cells (*Chávez-Galán et al., 2015*), is in fact expressed both in resident microglia and CCR2 recruited myeloid cells (*Figure 1f*), confirming data published elsewhere (*Valdearcos et al., 2019*). Next, to prepare the samples for RNA-sequencing, we sorted hypothalamic microglia expressing either CX3CR1 and recruited myeloid cells expressing CCR2 (*Figure 2a*). The quality of sorting was confirmed by determining the positivity for CX3CR1 (*Figure 2b*) and CCR2 (*Figure 2c*), and by determining the positivity for several markers of resident microglia (*Figure 2d–m*) and of bone marrow-derived cells (*Figure 2n–x*). The elucidation of the transcriptional landscapes of CX3CR1 and CCR2 cells was performed using bioinformatic tools to compare transcript expression levels in distinct cell types and conditions. The results (*Table 1*) revealed that either diet or sex exerted only minor differences in the expression of transcripts in CX3CR1 cells (*Figure 3a, b*); nevertheless, sex exerted major differences in CCR2 cells (*Figure 3a, b*). The direct comparisons between CX3CR1 and CCR2 obtained from mice fed on the HFD, revealed the vast differences in the transcriptional landscapes of either female (*Figure 3c*) or male (*Figure 3d*) mice. Furthermore, the direct comparisons between female and male mice revealed a considerable degree of sexual dimorphism in the transcriptional landscapes of recruited CCR2 cells (*Figure 3e*). In CX3CR1 cells, the consumption of the HFD impacted on IL17, lipids, toll-like receptor signaling, tumor necrosis factor signaling and chemokines (*Figure 4a*); whereas in CCR2 cells, the consumption of the HFD impacted on lipids, toll-like receptor signaling, tumor necrosis factor signaling, chemokines, neurotrophins signaling, reactive oxygen species, thermogenesis, and pathways related to neurodegeneration (*Figure 4a*). Next, we asked what functions related to chemotaxis were predominantly regulated in CCR2 cells of mice fed on the HFD (*Figure 4b, c*). As cell chemotaxis emerged as an important function in both females (*Figure 4b*) and males (*Figure 4c*), we looked with greater detail into the expression of chemokines and chemokine receptors (*Figure 4d*). Virtually all the transcripts evaluated showed diametrically opposite expression between CX3CR1 and CCR2 (*Figure 4d*).

### The impact of ovariectomy on diet-induced hypothalamic inflammation

Because sex exerted a major effect on the transcriptional signature of CCR2 cells, we evaluated ovariectomized mice fed on HFD, as depicted in the experimental design panel (*Figure 4—figure supplement 1a*). The ovariectomized mice presented increased body mass gain (*Figure 4—figure supplement 1b*) accompanied by increased adiposity (*Figure 4—figure supplement 1e*). This was accompanied by no change in food intake (*Figure 4—figure supplement 1c*) and blood glucose (*Figure 4—figure supplement 1d*). Regarding the hypothalamic expression of chemokines, there was only a reduction of *Cxcl16* (*Figure 4—figure supplement 1f*), and no change in the expression of chemokine receptors (*Figure 4—figure supplement 1g*). Regarding the hypothalamic expression of neuropeptides involved in energy balance, there was an increase in *Pomc* and a reduction in *Agrp* (*Figure 4—figure supplement 1h*). There were no changes in the hypothalamic transcripts of inflammatory markers (*Figure 4—figure supplement 1i*). The blood levels of estradiol were determined (*Figure 4—figure supplement 1j*).

### Cxcr3 is highly expressed in recruited peripheral myeloid cells

As we were particularly interested in identifying factors involved in the recruitment of CCR2 to the hypothalamus, we evaluated cytokine receptors with high expression in CCR2 and low expression in CX3CR1. As depicted in *Figure 5*, *Ccr3* (*Figure 5a*), *Ccr7* (*Figure 5b*), *Ccr8* (*Figure 5c*), *Cxcr2*

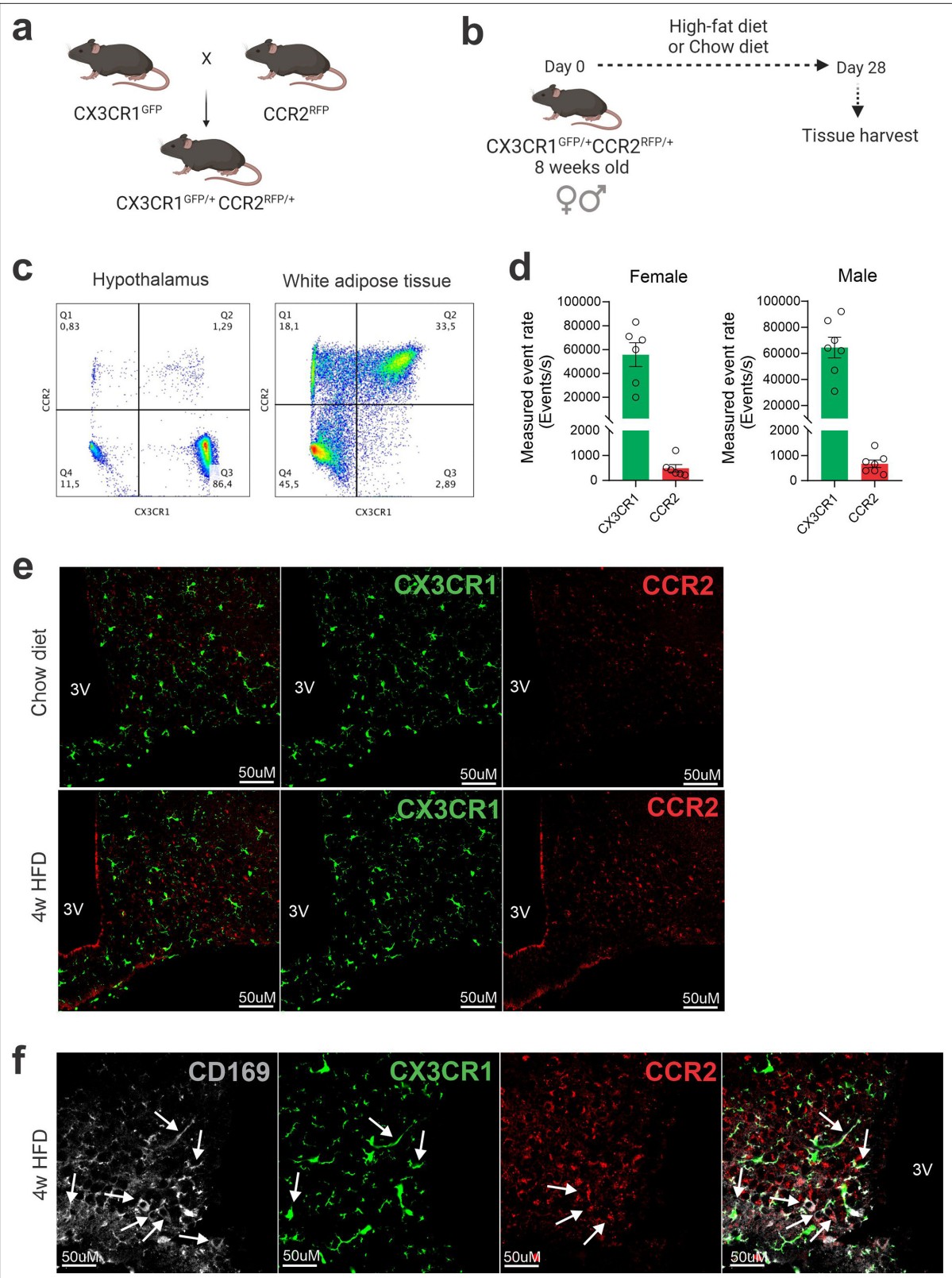

**Figure 1.** CCR2-positive cells infiltrate the hypothalamus of mice fed a high-fat diet (HFD). (**a**) CX3CR1$^{GFP/+}$CCR2$^{RFP/+}$ dual-reporter mutant mice generation. (**b**) Schematic representation of the experimental protocol for analysis of HFD-induced CCR2 peripheral-cell chemotaxis toward the hypothalamus. (**c**) Flow cytometry analysis of CX3CR1$^{GFP+}$ and CCR2$^{RFP+}$ cells in the white adipose tissue and in the hypothalamus of HFD-fed mice. (**d**) Measured event rate detected by flow cytometer of CX3CR1$^{GFP+}$ and CCR2$^{RFP+}$ cells isolated from the hypothalamus of HFD-fed male and female

*Figure 1 continued*

mice. (**e**) Coronal brain sections of mediobasal hypothalamus (MBH) from chow- and 4 weeks HFD-fed mice CX3CR1$^{GFP/+}$CCR2$^{RFP/+}$. (**f**) Coronal brain sections of MBH from 4 weeks HFD-fed mice CX3CR1$^{GFP/+}$CCR2$^{RFP/+}$ immunostained for CD169 (Sialoadhesin). White arrows indicate overlap between CD169-positive cells with CX3CR-positive cells or with CCR2-positive cells. 3V = third ventricle, scale bar = 50 μm.

---

(*Figure 5d*), *Cxcr3* (*Figure 5e*), *Cxcr4* (*Figure 5f*), *Cxcr5* (*Figure 5g*), *Cxcr6* (*Figure 5h*), and *Cxcr7* (*Figure 5i*) were all expressed in CCR2 cells and virtually absent from CX3CR1 cells. *Cxcr3* (*Figure 5e*) and *Cxcr6* (*Figure 5h*) presented the highest expressions, and therefore, we performed a search for previous studies looking at either of these chemokine receptors in the context of DIO hypothalamic inflammation. Using the terms, hypothalamus, hypothalamic, obesity, inflammation, *Cxcr3*, and *Cxcr6*, we could find no prior publications. As CXCR3 is involved in interferon-gamma (IFN-γ) induction (*Cole et al., 1998*), and IFN-γ has been shown to be expressed in the context of DIO hypothalamic inflammation (*De Souza et al., 2005*), we looked into the IFN-γ-related pathways regulated in recruited immune cells (*Figure 6*). First, we asked if the canonical ligands for CXCR3, *CXCL9*, CXCL10, and CXCL11 were expressed in either CX3CR1 or CCR2 cells. *Cxcl11* was not detected in either cell type (not shown). *Cxcl9* (*Figure 6a*) was expressed only in CX3CR1 cells, whereas *Cxcl10* (*Figure 6b*) was expressed in both CX3CR1 and CCR2 cells. In addition, in both female and male mice, *Ifng* was expressed in CCR2, but not in CX3CR1 cells (*Figure 6c*). Furthermore, IFN-γ pathways were shown to be modulated in both female (*Figure 6d*) and male (*Figure 6e*) CCR2 cells. Thus, we elected CXCR3 as a target for further intervention.

## The immunoneutralization of hypothalamic CXCL10 leads to increased body mass gain in female mice

As an attempt to interfere with CXCR3 actions in CCR2 cells, we targeted one of its ligands, CXCL10. As depicted in *Figure 7a*, mice were submitted to two intracerebroventricular (ICV) injections of an anti-CXCL10 antibody aimed at immunoneutralizing the target protein in the hypothalamus. As a result of the immunoneutralization of CXCL10, there were smaller numbers of CCR2-positive cells in the hypothalamus of both female and male mice fed on a HFD (*Figure 7b*). However, there were no major changes in the numbers of CXCR3-expressing cells in the hypothalamus of either female (*Figure 7c*) or male (*Figure 7d*) mice fed on HFD. This was accompanied by no changes in the transcript levels of *Cxcr3* and several other chemokine-related transcripts (*Figure 7c, d*), except for a trend to decrease *Cxcl1*1 and an increase of *Cxcr4* in females (*Figure 7c*); and a decrease of *Cxcl10* and a trend to increase *Cx3cl1* in males (*Figure 7d*). Nevertheless, the immunoneutralization of hypothalamic CXCL10 (*Figures 8 and 9*) resulted in increased body mass gain (*Figure 8a, b*), a trend to reduce blood triglycerides (*Figure 8f*), reduced blood cholesterol (*Figure 8g*), reduced expression of *Agrp* transcript in the hypothalamus (*Figure 8k*), a trend to reduce blood insulin (*Figure 8n*), trends to reduce *Il1b* and *Il6* transcripts in the hypothalamus (*Figure 8o*), and a trend to reduce respiratory quotient (*Figure 8s*) during the dark cycle in female mice. Conversely, in male mice (*Figure 9*), the inhibition of hypothalamic CXCL10 had only a minor effect, leading to a trend to reduce hypothalamic *Npy* (*Figure 9k*), a trend to reduce hypothalamic *Il1b*, and a trend to increase hypothalamic *Il6* (*Figure 9o*).

## The inhibition of CXCR3 worsens body mass gain and the metabolic phenotype of mice fed on a HFD

CXCR3 was inhibited using a pharmacological antagonist, AMG487 24 (*Figure 10a*). We decided to perform the treatment systemically, instead of ICV, because the purpose was to mitigate the migration of the cells expressing this receptor to the hypothalamus. Indeed, the intervention resulted in the reduction of CCR2 (*Figure 10b*) and CXCR3 (*Figure 10c, d*) cells in the hypothalamus of both female and male mice fed on HFD. In addition, there was a reduction of Ccl2 and an increase of Cx3cl1 transcripts in the hypothalamus of female (*Figure 10e*), and a reduction of Ccl2 transcripts in the hypothalamus of male (*Figure 10f*) mice fed on HFD. The inhibition of CXCR3 had a major impact on metabolic phenotype; thus, in female mice fed a HFD, there was an increase in body mass gain (*Figure 11a, b*), a trend to increase brown adipose tissue mass (*Figure 11c*), a trend to increase blood leptin (*Figure 11e*), an increase in blood triglycerides (*Figure 11f*), a trend to reduce hypothalamic

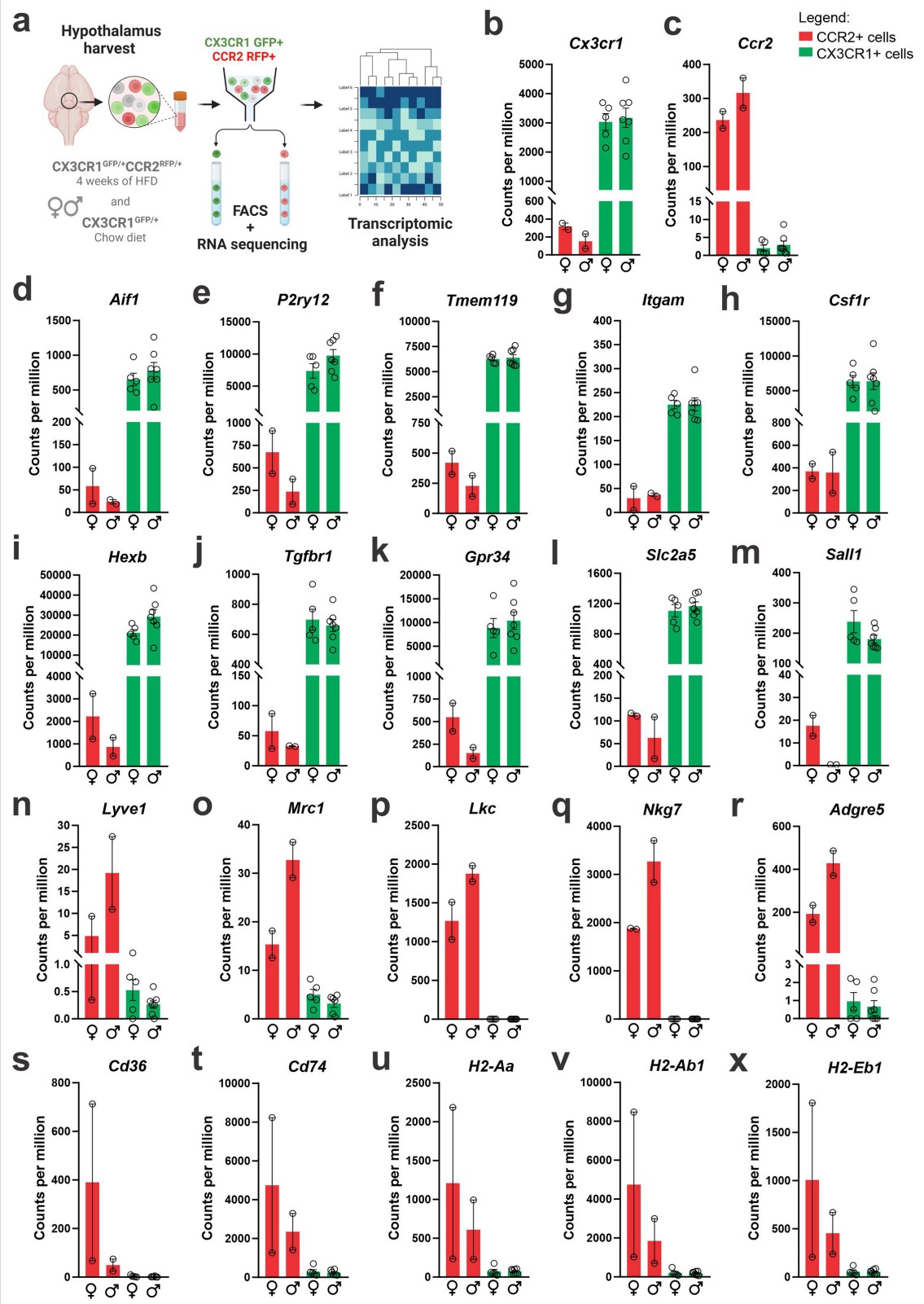

**Figure 2.** CX3CR1-positive resident microglia and CCR2-positive recruited myeloid cells sorted from the hypothalamus of high-fat diet (HFD)-fed mice express classical markers of microglia and other immune cells. (**a**) Schematic representation of the experimental protocol for sorting and sequencing CX3CR1 and CCR2 cells from the hypothalamus of chow- and HFD-fed mice. (**b**) *Cx3cr1* gene expression and (**c**) *Ccr2* gene expression of CX3CR1 and CCR2 cells sorted from the hypothalamus of HFD-fed mice. Analysis of (**d–m**) classical microglial markers, and (**n–x**) bone marrow-derived immune

*Figure 2 continued on next page*

*Figure 2 continued*

cell markers in the transcriptome of CX3CR1 and CCR2 cells sorted from the hypothalamus of HFD-fed mice. To perform RNA-sequencing (RNA-seq) we have employed a total of 200 mice of each sex fed on HFD and 100 mice of each sex fed on chow diet. They were divided into five independent experiments. To get the total RNA amount from CCR2-positive cells in the hypothalamus of HFD-fed mice that was enough for library construction and RNA-seq, CCR2 samples were pooled together in three samples, but only two samples could be sequenced due to the final RNA integrity and amount.

Pomc (*Figure 11k*), an increase in hypothalamic Npy (*Figure 11k*), a worsen glucose tolerance (*Figure 11l*), an increased fasting blood glucose (*Figure 11m*), a reduced blood insulin (*Figure 11n*), and reduction of Il6 and Tlr4 transcripts in the hypothalamus (*Figure 11o*). In males, the inhibition of CXCR3 promoted an increased body mass gain (*Figure 12a, b*), increased WAT mass (*Figure 12d*), increased blood leptin (*Figure 12e*), increased hypothalamic Npy and Mch (*Figure 12k*), and reduced hypothalamic Tnfa and Nlrp3 (*Figure 12*). Additionally, the inhibition of CXCR3 promoted changes in neither energy expenditure nor locomotor activity (*Figure 11—figure supplement 1a, b*, for females and *Figure 12—figure supplement 1a, b* for males). In the hypothalamus of females there were no changes in the expression of transcripts encoding proteins involved in endoplasmic reticulum homeostasis (*Figure 11—figure supplement 1c*) and mitochondrial turnover (*Figure 11—figure supplement 1d*), whereas in males there was a reduction of Ddit3 (*Figure 12—figure supplement 1c*) and Mfn1 (*Figure 12—figure supplement 1d*). Moreover, in females the inhibition of CXCR3 promoted no changes in the liver expression of lipidogenic and gluconeogenic genes (*Figure 11—figure supplement 1e*), and no changes in the WAT expression of lipidogenic genes (*Figure 11—figure supplement 1f*). In the liver of males, there was a reduction in the expression of Fasn and an increase in the expression of G6pc3 (*Figure 12—figure supplement 1e*). As for the females, in males, there were no changes in the WAT expression of lipidogenic genes (*Figure 12—figure supplement 1f*).

## Discussion

In this study, we elucidated the transcriptional landscapes of resident microglia and recruited myeloid cells in the hypothalamus of mice. We showed that hypothalamic microglia undergo minor transcriptional changes when mice are fed on HFD; however, there are vast differences when confronting resident CX3CR1 microglia versus recruited CCR2 myeloid cells transcriptomes. Moreover, there is a considerable degree of sexual dimorphism in the transcriptomes of recruited CCR2 myeloid cells

**Table 1.** Number of differentially expressed genes (DEGs) for each comparison.

|  |  | ↑ Up | ↓ Down |
|---|---|---|---|
|  | hfd_cx_f<br>Cx3cr1_hfd_female vs.<br>Cx3cr1_chow_female | 25 | 9 |
| Effect of the diet in brain resident microglia cells | hfd_cx_m<br>Cx3cr1_hfd_male vs.<br>Cx3cr1_chow_male | 261 | 151 |
|  | chow_cx<br>Cx3cr1_chow_male vs.<br>Cx3cr1_chow_female | 9 | 11 |
| Effect of the sex in brain resident microglia cells | hfd_cx<br>Cx3cr1_hfd_male vs.<br>Cx3cr1_hfd_female | 7 | 5 |
|  | hfd_cc_f<br>Cx3cr1_hfd_female vs.<br>Ccr2_hfd_female | 4036 | 3569 |
| Brain resident microglia cells vs. recruited myeloid cells | hfd_cc_m<br>Cx3cr1_hfd_male vs.<br>Ccr2_hfd_male | 3838 | 3350 |
| Effect of the sex in recruited myeloid cells | hfd_cc<br>Ccr2_hfd_male vs.<br>Ccr2_hfd_female | 1598 | 1676 |

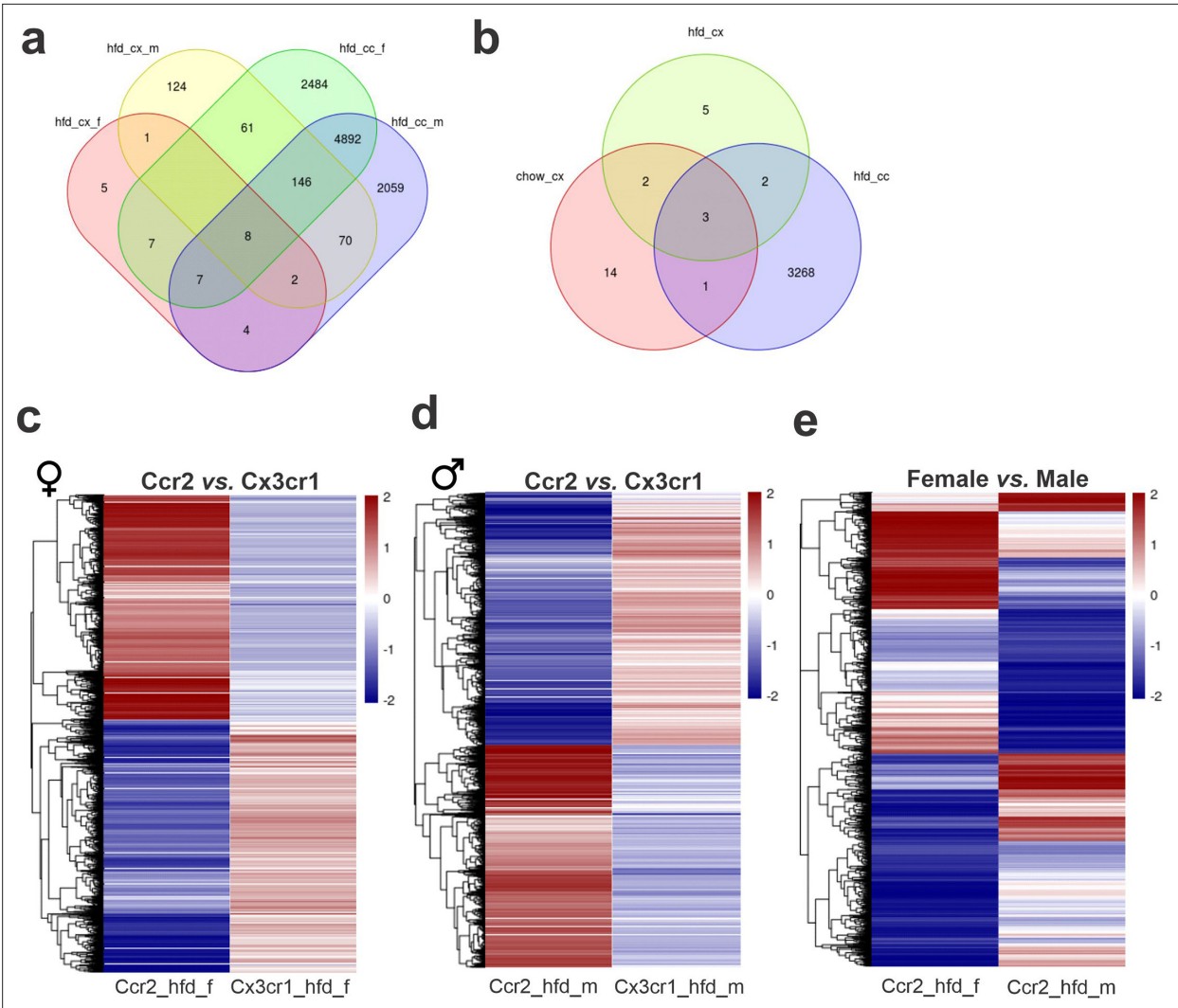

**Figure 3.** Differential gene expression (DGE) analysis of CX3CR1-positive resident microglia and CCR2-positive recruited myeloid cells sorted from the hypothalamus of high-fat diet (HFD)-fed mice show an enormous difference in their transcriptomic signature. (**a, b**) Venn diagram showing the number of DGEs for chow and HFD diet comparison and sex comparison, respectively. (**c**) Heatmap of up- and downregulated DGEs when comparing CX3CR1 and CCR2 cells from HFD-fed female mice. (**d**) Heatmap of up- and downregulated DGEs when comparing CX3CR1 and CCR2 cells from HFD-fed male mice. (**e**) Heatmap of up- and downregulated DGEs when comparing CCR2 recruited myeloid cells from HFD-fed male and female mice. Legend: hfd_cx_f = Cx3cr1_hfd_female vs. Cx3cr1_chow_female; hfd_cx_m = Cx3cr1_hfd_male vs. Cx3cr1_chow_male; chow_cx = Cx3cr1_chow_male vs. Cx3cr1_chow_female; hfd_cx = Cx3cr1_hfd_male vs. Cx3cr1_hfd_female; hfd_cc_f = Cx3cr1_hfd_female vs. Ccr2_hfd_female; hfd_cc_m = Cx3cr1_hfd_male vs. Ccr2_hfd_male; hfd_cc = Ccr2_hfd_male vs. Ccr2_hfd_female.

in mice fed a HFD. Upon exploration of the differences between resident microglia and recruited immune cells, we identified the chemokine receptor CXCR3 as an interesting candidate for intervention as it was highly expressed in recruited cells. The inhibition of CXCR3 resulted in increased body mass gain, worsening of glucose intolerance, and increased mRNA expression of hypothalamic *Npy*. Thus, the study is the first to identify a subset of recruited myeloid cells that has a protective role against the deleterious outcomes of DIO, therefore establishing a new concept in obesity-associated hypothalamic inflammation.

Early studies in this field have shown that dietary fats, particularly long-chain saturated fatty acids, trigger an inflammatory response in the mediobasal hypothalamus that emerges a few hours after the introduction of a HFD and progresses to chronicity if the consumption of the HFD persists for long (*De Souza et al., 2005*; *Milanski et al., 2009*; *Zhang et al., 2008*). Microglia play an important role in this inflammatory response, and studies have shown that during the course of a prolonged consumption of HFD, there is the recruitment of bone marrow-derived cells to compose a new hypothalamic microglia

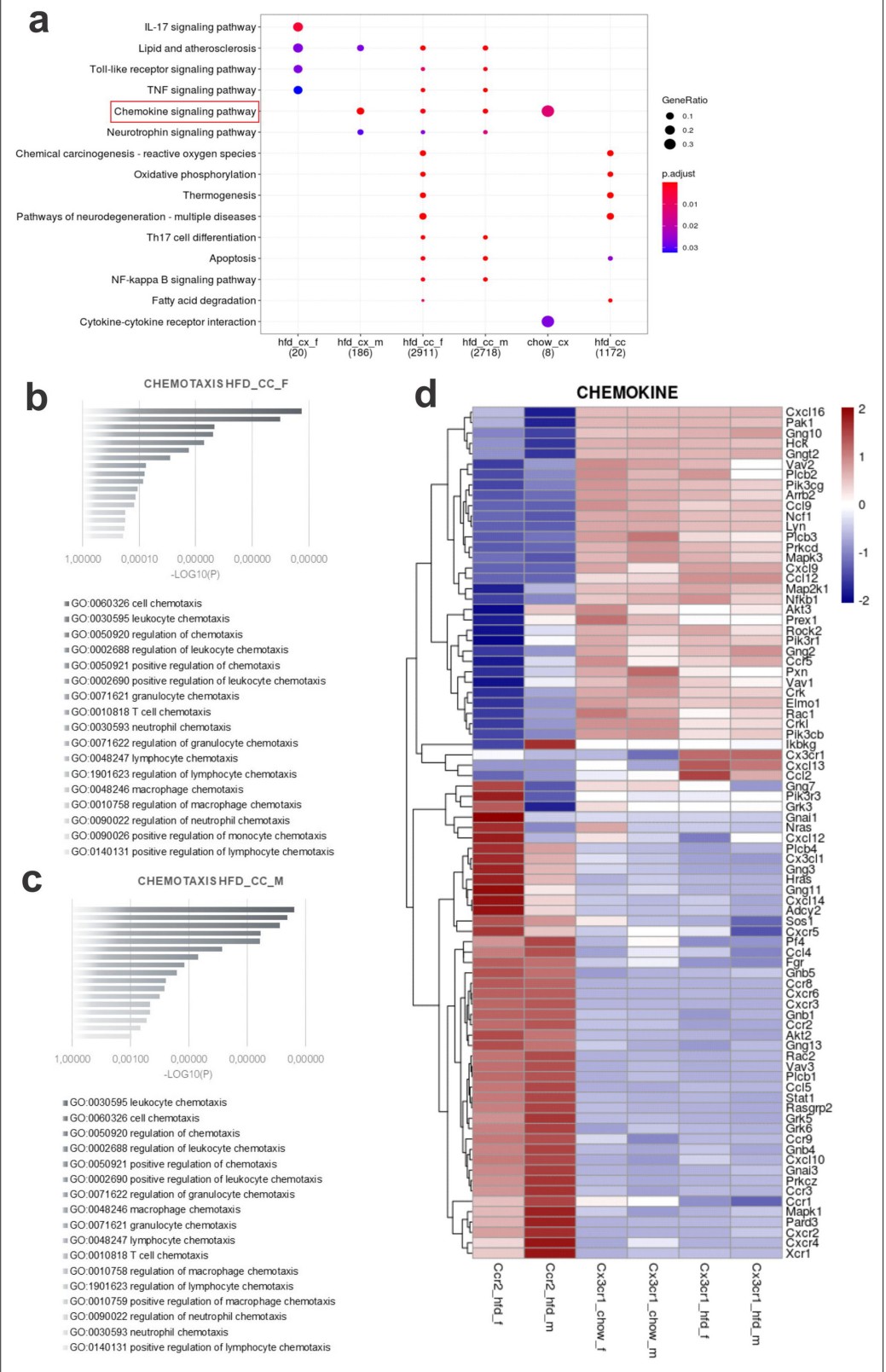

**Figure 4.** Various differential gene expression (DGE) found in CCR2+ infiltrating cells from the hypothalamus of high-fat diet (HFD)-fed mice belong to chemotaxis pathways. (**a**) Kioto Encyclopedia of Genes and Genomes (KEGG) enrichment analysis shows the distribution of DGEs in distinct metabolic pathways. (**b, c**) Ontology analysis for DGEs related to chemotaxis from CCR2-positive cells sorted from the hypothalamus of HFD-fed female and

*Figure 4 continued on next page*

*Figure 4 continued*

male mice, respectively. (**d**) Heatmap of up- and downregulated DGEs related to chemotaxis when comparing CX3CR1-positive microglia and CCR2-positive recruited myeloid cells from HFD-fed male and female mice.

The online version of this article includes the following figure supplement(s) for figure 4:

**Figure supplement 1.** Ovariectomy and ovariectomy with estradiol replacement modulate hypothalamic chemokine and neuropeptides.

landscape (*Thaler et al., 2012*; *Morari et al., 2014*; *Valdearcos et al., 2019*). However, it was previously unknown what are the transcriptional signatures of hypothalamic microglia and recruited bone marrow-derived cells in DIO.

To elucidate the transcriptional landscapes of hypothalamic microglia and recruited myeloid cells, we initially prepared a dual-reporter mouse for CX3CR1 and CCR2. CX3CR1 encodes for the fractalkine receptor and is highly expressed in resident microglia (*Greter et al., 2015*). The creation of CX3CR1 reporter mice was regarded as an important step toward the characterization of resident microglia, and several studies in the field employ this model (*Goldmann et al., 2013*; *Yona et al., 2013*). Conversely, CCR2 is highly expressed in bone marrow-derived cells (*Greter et al., 2015*; *Mildner et al., 2007*). The quality of our model was proven good as markers of resident microglia were present only in CX3CR1 cells, whereas markers of recruited myeloid cells were present only in CCR2 cells. Moreover, as previously described, we could find no CCR2 cells in the hypothalamus of mice fed chow (*Valdearcos et al., 2017*; *Valdearcos et al., 2019*).

The first important, and previously unknown finding emerged from the comparison of the transcriptomes of hypothalamic microglia in mice fed chow versus mice fed HFD. In females, there were only 34 transcripts undergoing significant changes between the two dietary conditions, whereas in males, the number was greater, 412, but still quite small considering the whole transcriptome of resident microglia (*Young et al., 2021*). In a study evaluating the single-cell transcriptomics of hypothalamic cells, the consumption of a HFD resulted in minor changes in the so-called macrophage-like

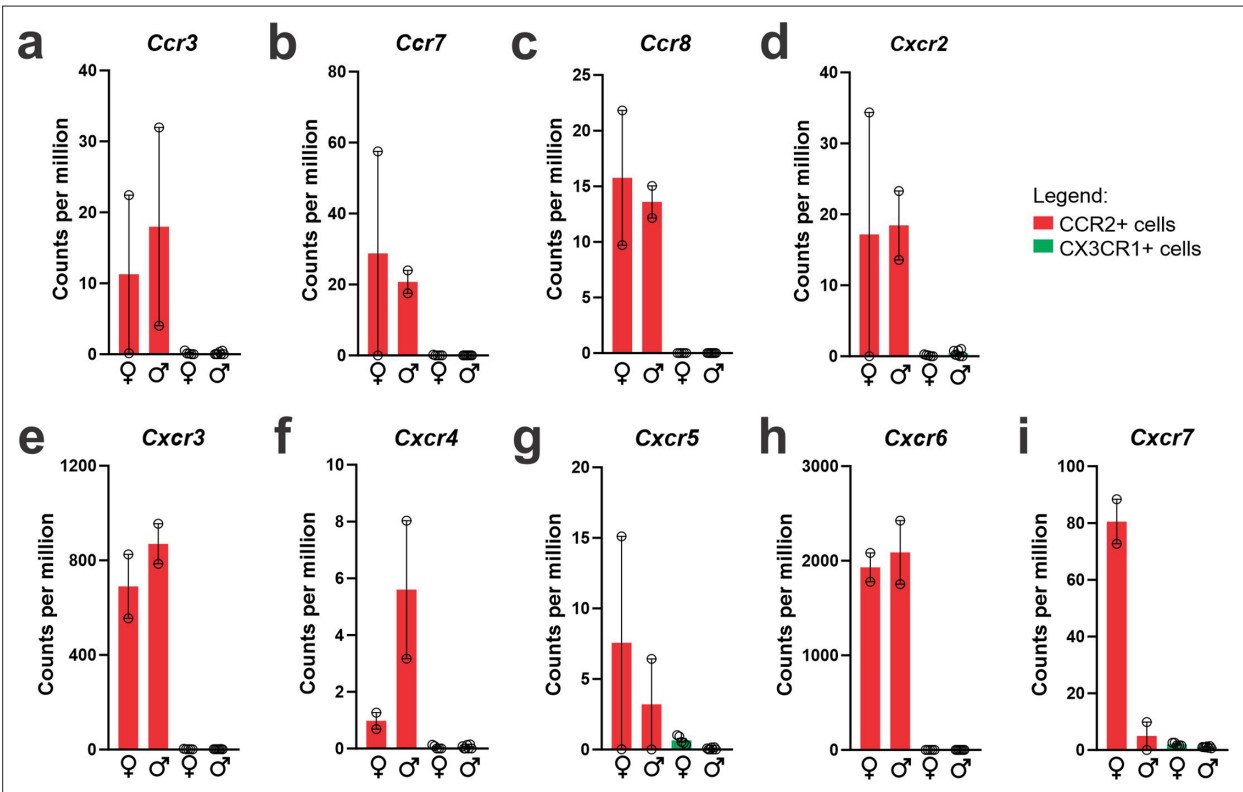

**Figure 5.** CCR2-positive recruited myeloid cells from the hypothalamus of high-fat diet (HFD)-fed mice express a broad range of chemokine receptors. (**a–i**) Chemokine receptors gene expression in the transcriptome of CX3CR1- and CCR2-positive cells sorted from the hypothalamus of HFD-fed mice.

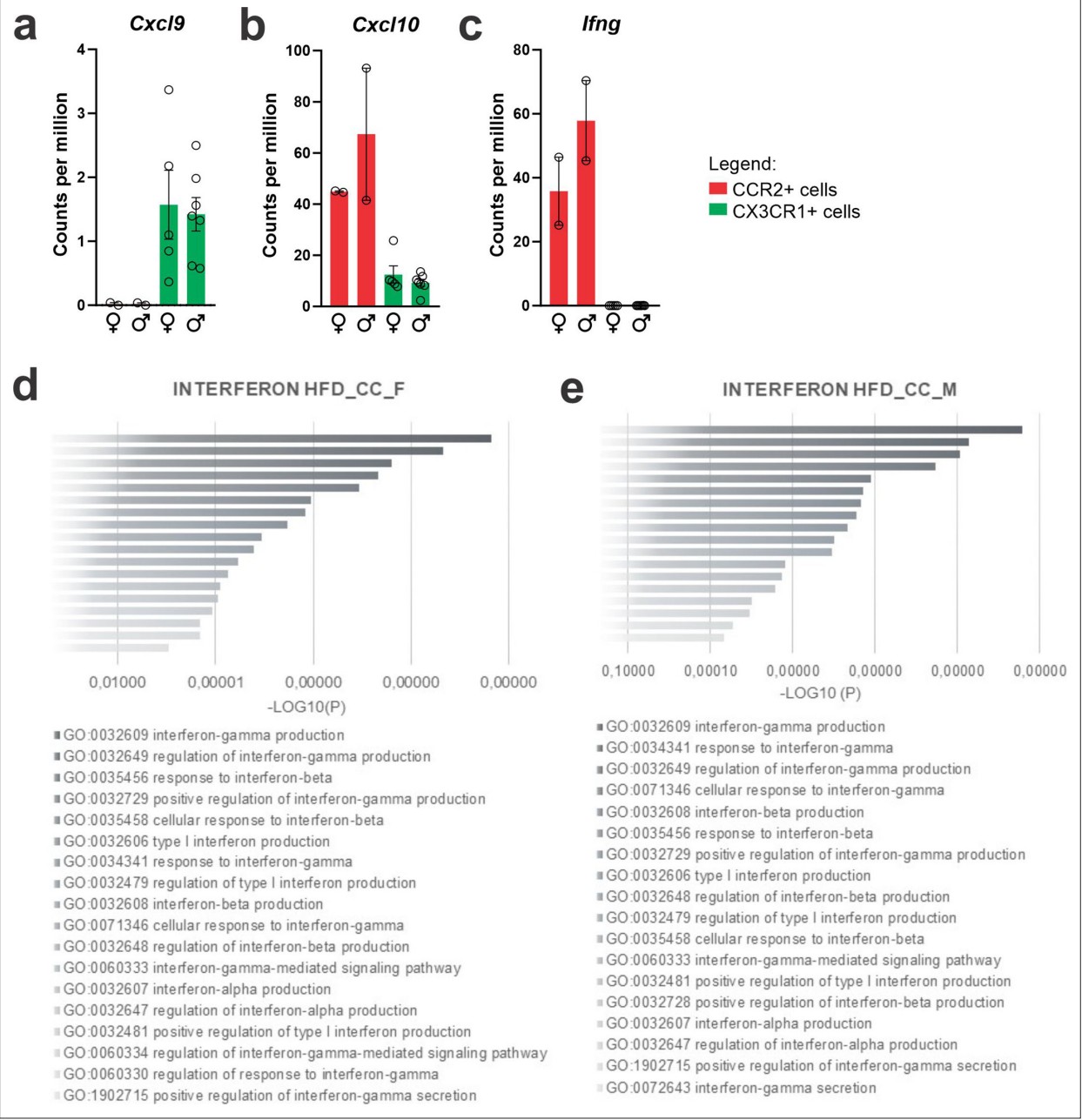

**Figure 6.** CXCL10/interferon γ-induced protein 10 kDa (IP-10) is highly expressed in CCR2-poisitve recruited myeloid cells from the hypothalamus of high-fat diet (HFD)-fed mice. (**a–c**) *Cxcl9*, *Cxcl10*, and *Ifng* gene expression in the transcriptome of CX3CR1- and CCR2-positive cells sorted from the hypothalamus of HFD-fed mice. (**d, e**) Ontology analysis for DGEs related to interferon signaling pathways from CCR2-positive cells sorted from the hypothalamus of HFD-fed female and male mice, respectively.

cells (***Campbell et al., 2017***); however, the detailed transcriptional landscape of these cells was not explored in depth, so it is uncertain if it contained both resident microglia and recruited immune cells. In an experimental model of Alzheimer's disease, which evaluated male and female mice, there were hippocampal resident microglial transcriptional changes of the same magnitude as the one we found in the hypothalamus, affecting approximately 300 genes (***Rivera-Escalera et al., 2019***). In an experimental model of cerebral hemorrhage evaluating only males, the impact on microglia transcriptome was also small, affecting only 10% of the evaluated genes (***Taylor et al., 2017***). In addition, in a study evaluating transcriptional changes of resident microglia during aging (***Li et al., 2023***), there were also important differences between female and male mice; and, the magnitude of the

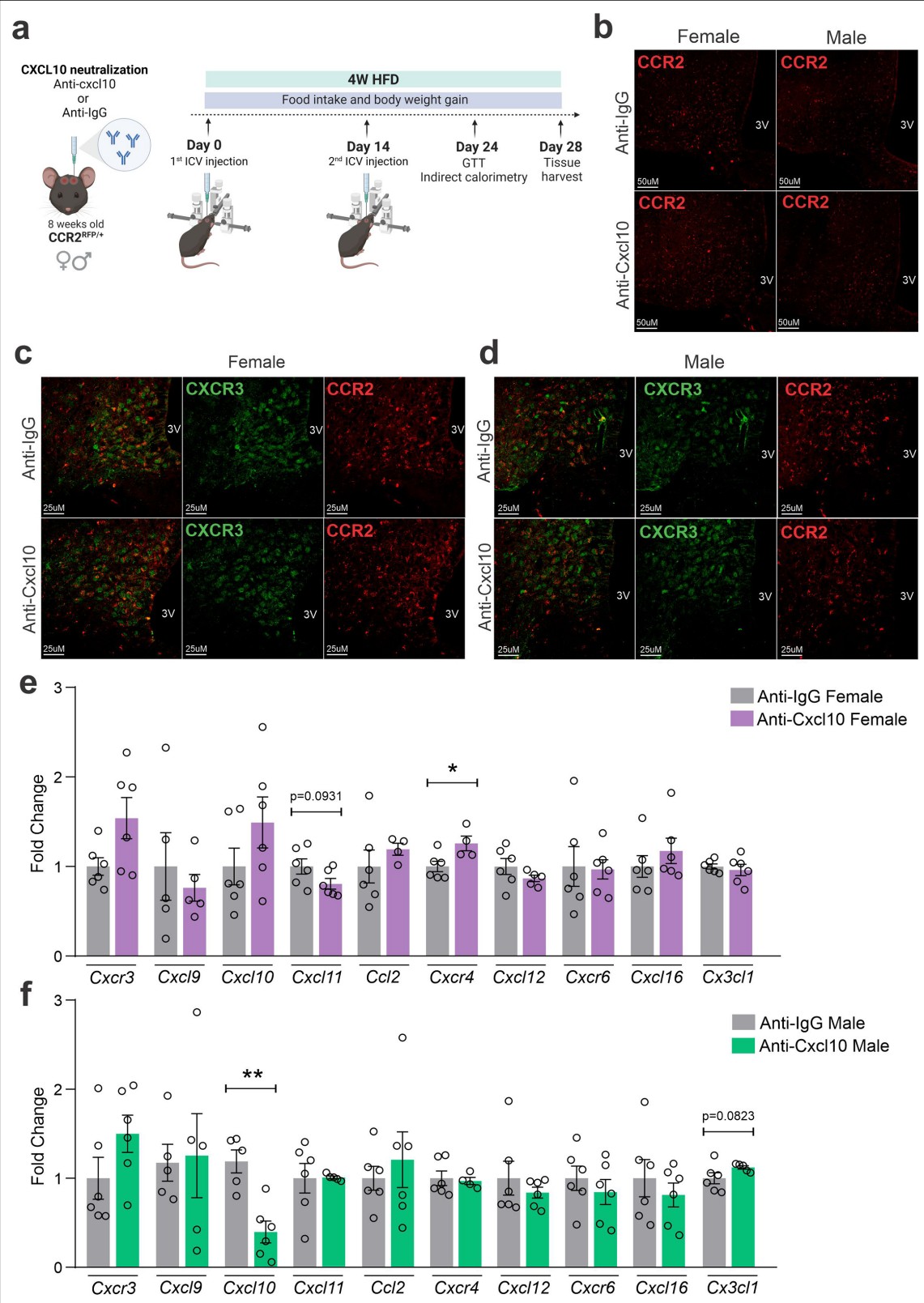

**Figure 7.** CXCL10 neutralization has a mild impact on reducing CCR2- and CXCR3-positive cell chemotaxis toward the hypothalamus of high-fat diet (HFD)-fed mice. (**a**) Schematic representation of the experimental protocol for CXCL10 central neutralization. (**b**) Coronal brain sections from 4 weeks HFD-fed CCR2[RFP/+] mice showing the CCR2-positive cells distribution in the hypothalamic parenchyma upon CXCL10 central neutralization. 3V = third ventricle, scale bars = 50 μm. (**c**) Coronal brain sections from 4 weeks HFD-fed female CCR2[RFP/+] mice immunostained for CXCR3 upon CXCL10 central

*Figure 7 continued on next page*

Figure 7 continued

neutralization. (**d**) Coronal brain sections from 4 weeks HFD-fed male CCR2[RFP/+] mice immunostained for CXCR3 upon CXCL10 central neutralization. 3V = third ventricle, scale bars = 25 µm. (**e, f**) Hypothalamic mRNA levels of several chemokine receptors and chemokines in HFD-fed female (light gray and purple bars) and male (light gray and green bars) CCR2[RFP/+] mice upon CXCL10 central neutralization. For qualitative confocal image analysis, we have used 3 samples per group. For real-time quantitavive polymerase chain reaction (RT-qPCR) of the hypothalamus, we have used 5–6 samples per group. Two-tailed Mann–Whitney tests were used for statistical analyses. *p < 0.05 and **p < 0.01 in comparison with respective Anti-IgG-treated groups.

transcriptional changes occurring during aging was about the same as we see in the dietary intervention. Thus, it seems that in different brain regions and under distinct interventions, the magnitude of resident microglia transcriptional changes is quite small; however, the number of studies evaluating this question is not expressive; thus, further studies are needed to provide a definitive view regarding the actual magnitude of plasticity of resident microglia. Despite the small number of transcripts undergoing changes in our model, greatest impact occurred in the expression of genes related to IL-17 signaling, lipid metabolism, toll-like receptor signaling, tumor necrosis factor signaling and chemokine signaling, which strongly supports the role of dietary lipids in the induction of an inflammatory response by the resident microglia, supporting previous studies in the field (*Milanski et al., 2009*; *Thaler et al., 2012*; *Morari et al., 2014*; *Valdearcos et al., 2019*). Interestingly, IL17 signaling emerged as a major pathway modulated by DIO. In a recent study, we have shown that IL17 can act upon proopiomelanocortin (POMC) neurons promoting a reduction in calorie intake (*Nogueira et al., 2020*). The current finding of a transcriptional modulation of IL17-related genes in resident microglia opens a new perspective in the understanding of how inflammatory signals modulate the function of the hypothalamus in obesity, which should be explored in the future.

Next, we confronted the transcriptomes of hypothalamic resident microglia and recruited myeloid cells, and in this case, there were huge differences, reaching over 7000 transcripts. Per se, this finding reveals a striking difference between resident microglia and recruited immune cells in the hypothalamus of mice fed on HFD. Nevertheless, despite this is a new information regarding the hypothalamus, similar findings were reported in other brain regions submitted to distinct interventions, such as in an experimental model of glioma (*Ochocka et al., 2021*), in flavivirus infection (*Spiteri et al., 2023*), and in COVID-19 (*Schwabenland et al., 2021*; *Hartmann et al., 2024*). The evaluation of the main pathways differentially expressed in the two cell subsets revealed major differences in lipids, toll-like receptor signaling, tumor necrosis factor signaling, chemokines, neurotrophins signaling, reactive oxygen species, thermogenesis, and pathways related to neurodegeneration. The impact of the consumption of a HFD on the regulation of lipid-related pathways, toll-like receptor, and tumor necrosis factor signaling has been widely explored in many previous studies, and interventions in these systems are known to mitigate the harmful effects of the diet (*Milanski et al., 2009*; *Thaler et al., 2012*; *Morari et al., 2014*; *Valdearcos et al., 2019*). However, little has been done regarding the characterization of the mechanisms of chemotaxis that drive the recruitment of bone marrow-derived cells to the hypothalamus. Therefore, we looked with greater detail into the main chemokines and chemokine receptors differentially expressed in the two subsets of cells. We elected CXCR3 because it was highly expressed in CCR2 cells and presented low expression in CX3CR1 cells. CXCR3 is a chemokine receptor that is involved in the recruitment of distinct types of bone marrow-derived monocytic cells, such as plasmacytoid monocytes (*Cella et al., 1999*), synovial tissue monocytes (*Katschke et al., 2001*), and dendritic cells (*Padovan et al., 2002*). In the brain, CXCR3-expressing cells have been implicated in Alzheimer's disease and other age-dependent cognitive dysfunctions (*Jorfi et al., 2023*; *Schroer et al., 2023*), multiple sclerosis (*Bogers et al., 2023*), epilepsy (*Liang et al., 2023*), and stroke (*Cai et al., 2022*). However, no previous study has evaluated CXCR3 in the hypothalamus in the context of obesity.

First, we asked if the known ligands for CXCR3, and *Ifng*, which is induced in response to the activation of CXCR3, were present in either subset of cells. We found *Cxcl11* in neither cell subset; *Cxcl9* was expressed only in CX3CR1 microglia; *Cxcl10* was expressed in both cell types, with greater expression in CCR2 recruited cells; and Ifng was expressed only in CCR2 cells. Moreover, there was a considerable engagement of IFN-γ-related pathways in CCR2 cells of both female and male mice fed a HFD. Thus, we considered CXCR3 as a promising target for intervention.

Next, we intervened in one of the ligands for CXCR3, CXCL10. For that, we performed ICV injections of an immunoneutralizing antibody, which resulted in smaller numbers of CCR2 cells in the

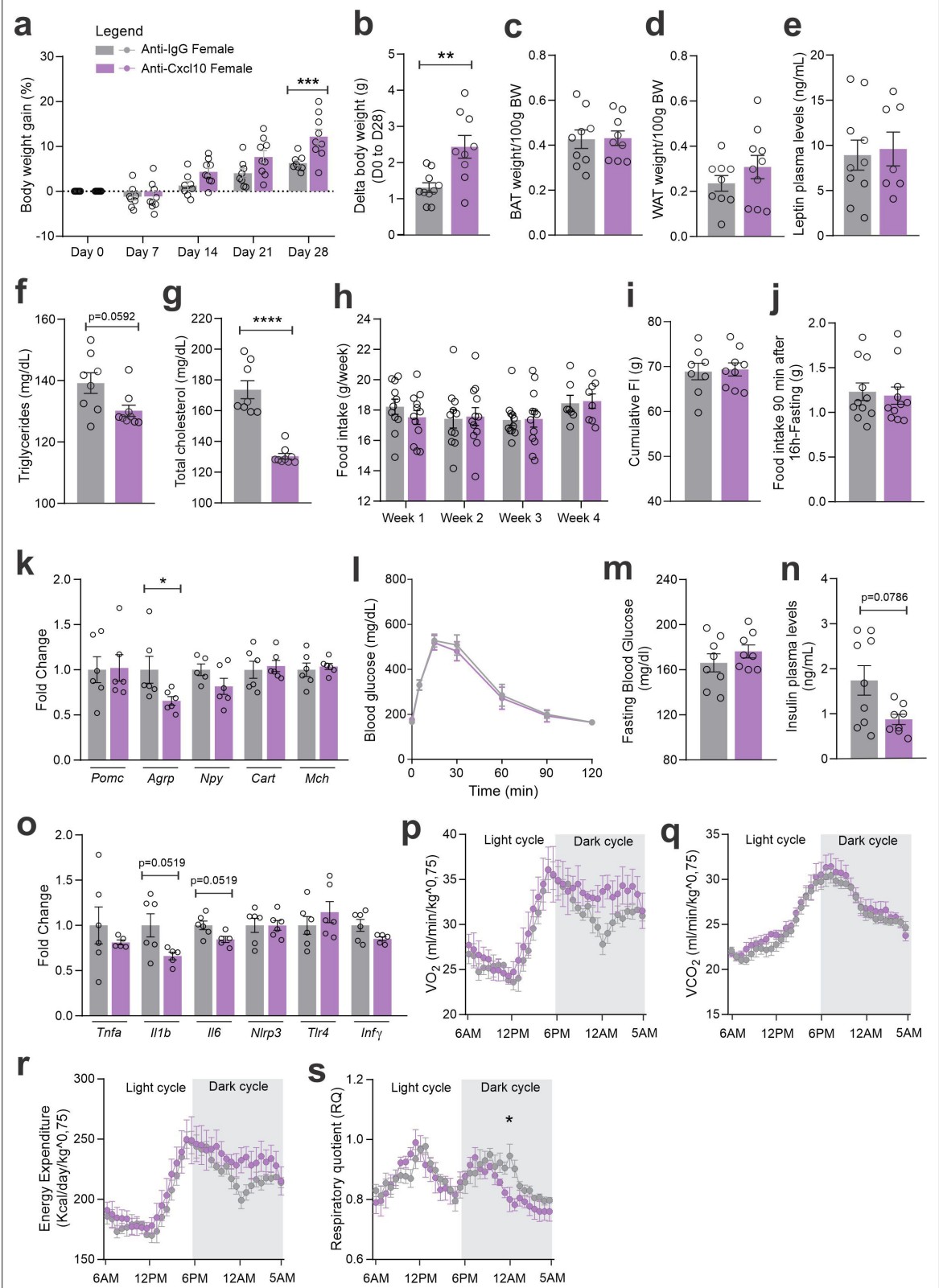

**Figure 8.** CXCL10 central neutralization in high-fat diet (HFD)-fed female mice. (**a**) Percentual of body weight gain from Day 0 to 28 of the experimental protocol. (**b**) Delta body weight during the experimental period. (**c**) Brown adipose tissue weight and (**d**) white adipose tissue (retroperitoneal depot) weight at Day 28. (**e**) Leptin, (**f**) triglycerides, and (**g**) total cholesterol plasma levels at Day 28. (**h**) Weekly food intake measurement during the experimental period. (**i**) Cumulative food intake during the experimental period. (**j**) 90-min food intake measurement after 16 hr of fasting.

*Figure 8 continued on next page*

*Figure 8 continued*

(**k**) Hypothalamic mRNA levels of neuropeptides involved in food intake control. (**l**) Intraperitoneal glucose tolerance test on Day 24. (**m**) 6-hr fasting blood glucose levels. (**n**) Insulin plasma levels at Day 28. (**o**) Hypothalamic mRNA levels of inflammatory genes. (**p**) $O_2$ consumption; (**q**) $CO_2$ production; (**r**) energy expenditure; (**s**) respiratory quotient at Day 24. Data were expressed as mean ± SEM of 8–10 mice/group (in two independent experiments). To perform quantitative reverse transcription-polymerase chain reaction (qRT-PCR) we have used 6 mice/group. To perform biochemical analysis in plasma we have used 8–10 mice/group. To perform ipGTT we have used 4 mice/group. To perform indirect calorimetry, we have used 4 mice/group. Two-way ANOVA followed by Sidak's post hoc test and Mann–Whitney test were used for statistical analyses. $*p < 0.05$, $**p < 0.01$, $***p < 0.001$, $****p < 0.0001$ in comparison with IgG-treated group.

hypothalamus. However, the intervention promoted minimal changes in the hypothalamic expression of transcripts encoding for several components of the chemotaxis machinery. Nevertheless, in females, the inhibition of CXCL10 resulted in increased body mass gain, reduction of hypothalamic *Agrp*, trends to reduce hypothalamic *Il1b* and *Il6*, and a trend to reduce blood insulin. In males, the phenotype was much milder, leading to minimal changes in hypothalamic *Npy*, *Il1b*, and *Il6*. Little is known about the involvement of CXCL10 in hypothalamic physiology and pathology. In a model of caloric restriction, there was an increase in the hypothalamic expression of CXCL10 (*Matthews et al., 2017*), and this was regarded as a component of the mechanism of neuroprotection induced by caloric restriction. In addition, in a model of hypothalamic inflammation elicited by exogenous LPS, CXCL10 emerged as one of the transcripts undergoing the greatest increase in the hypothalamic paraventricular nucleus (*Reyes et al., 2003*).

As CXCR3 can be engaged by distinct chemokines, we decided to use a broader intervention, pharmacologically inhibiting CXCR3. AMG487 is a chemical inhibitor of CXCR3 with an $IC_{50}$ value of 8.0 nM (*Johnson et al., 2007*). Upon treatment with AMG487, there was a reduction of the migration of CCR2 cells to the hypothalamus, which was accompanied by minimal changes in the expressions of chemokines and chemokine receptors. Under inhibition of CXCR3, both female and male mice presented increased body mass gain, which was accompanied by increased blood leptin, increased fasting glucose, increased hypothalamic *Npy* transcript, and reduction in markers of hypothalamic inflammation. There were no changes in caloric intake, however, there were reductions in energy expenditure during some periods during the 24 hr of recording.

These findings reveal that, at least one subset of recruited myeloid cells, has a protective rather than a harmful role in DIO-associated hypothalamic inflammation. This is a completely new concept in the field because all previous studies evaluating hypothalamic microglia in DIO reported that, once active in response to dietary fats, either resident microglia or recruited immune cells exerted inflammatory actions that impacted negatively energy balance and glucose tolerance (*Tapia-González et al., 2011*; *Valdearcos et al., 2014*; *Fernández-Arjona et al., 2022*; *Morari et al., 2014*). This concept has been recently explored in depth in a study that used elegant models to either activate or inactivate microglia (*Valdearcos et al., 2017*) and the results confirm that, whenever manipulating microglia using approaches that are not specific for a given subset of cells, the net result is worsening of the metabolic phenotype when microglia is activated and improvement of the metabolic phenotype when microglia is inactivated. Thus, we believe that the subset of recruited myeloid cells herein identified plays a regulatory role in hypothalamic inflammation.

We acknowledge that the study could be strengthened if, in addition to pharmacological and antibody-based interventions, we used gene-targeted approaches as well.

In conclusion, this study elucidated the transcriptional landscapes of hypothalamic resident microglia and recruited myeloid cells in DIO. In resident microglia, the consumption of a HFD resulted in small changes in transcript expression, whereas the confrontation of the transcriptional landscapes of resident microglia versus recruited immune cells revealed broad differences that encompass lipids, toll-like receptor signaling, tumor necrosis factor signaling, chemokines, neurotrophins signaling, reactive oxygen species, thermogenesis, and pathways related to neurodegeneration. In addition, the study revealed a considerable sexual dimorphism in the transcriptional landscape of both hypothalamic resident microglia and recruited immune cells. The study also identified a subset of recruited immune cells, expressing the chemokine receptor CXCR3 that has a protective role against the harmful metabolic effects of the HFD; thus, providing a new concept in DIO-associated hypothalamic inflammation.

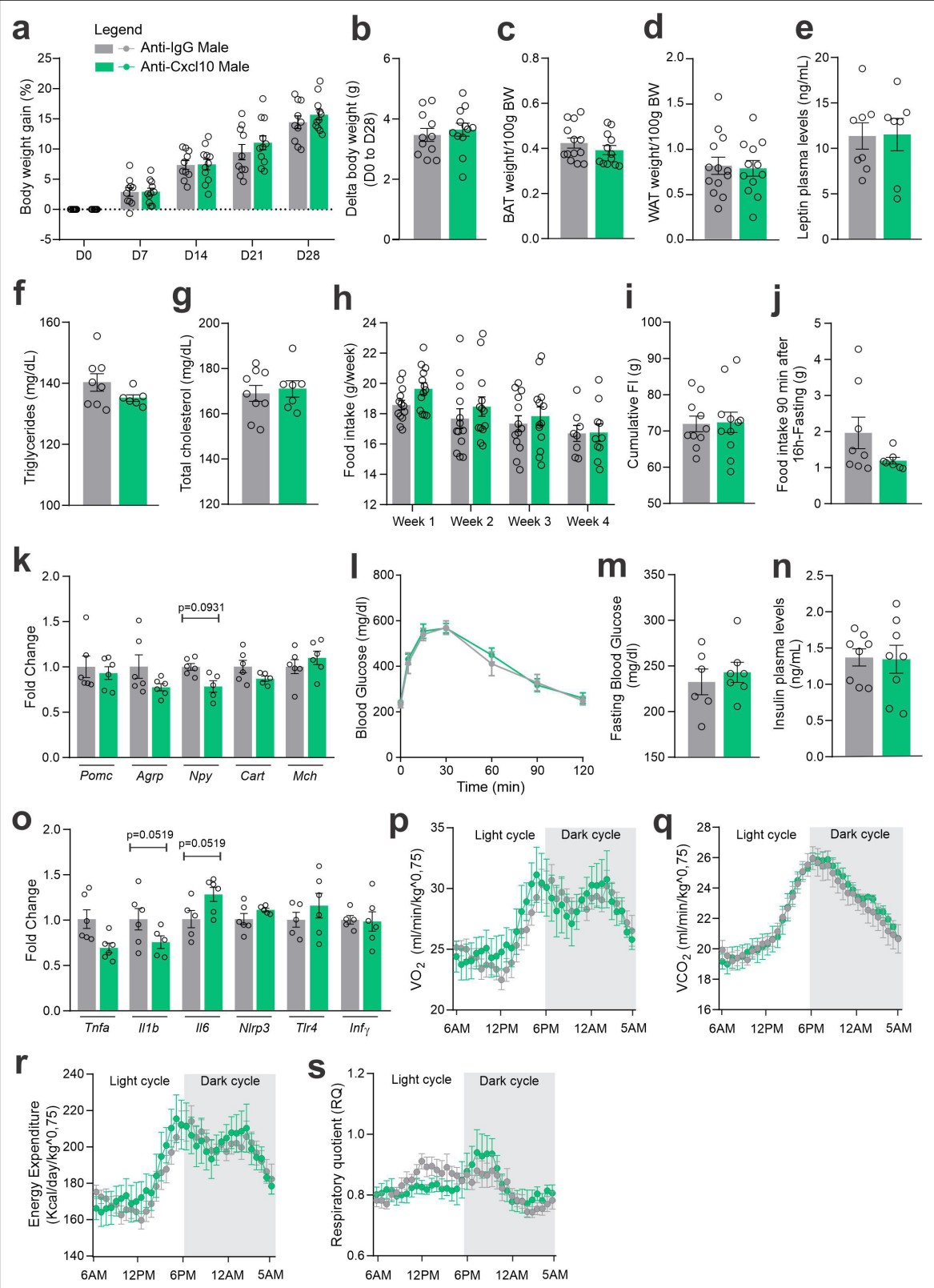

**Figure 9.** CXCL10 central neutralization in high-fat diet (HFD)-fed male mice. (**a**) Percentual of body weight gain from Day 0 to 28 of the experimental protocol. (**b**) Delta body weight during the experimental period. (**c**) Brown adipose tissue weight and (**d**) white adipose tissue (retroperitoneal depot) weight at Day 28. (**e**) Leptin, (**f**) triglycerides, and (**g**) total cholesterol plasma levels at Day 28. (**h**) Weekly food intake measurement during the experimental period. (**i**) Cumulative food intake during the experimental period. (**j**) 90-min food intake measurement after 16 hr of fasting.

*Figure 9 continued on next page*

*Figure 9 continued*

(**k**) Hypothalamic mRNA levels of neuropeptides involved in food intake control. (**l**) Intraperitoneal glucose tolerance test at Day 24. (**m**) 6-hr fasting blood glucose levels. (**n**) Insulin plasma levels at Day 28. (**o**) Hypothalamic mRNA levels of inflammatory genes. (**p**) $O_2$ consumption; (**q**) $CO_2$ production; (**r**) energy expenditure; (**s**) respiratory quotient at Day 24. Data were expressed as mean ± SEM of 8–10 mice/group (in two independent experiments). To perform quantitative reverse transcription-polymerase chain reaction (qRT-PCR) we have used 6 mice/group. To perform biochemical analysis in plasma we have used 8–10 mice/group. To perform ipGTT we have used 4 mice/group. To perform indirect calorimetry, we have used 4 mice/group. Two-way ANOVA followed by Sidak's post hoc test and Mann–Whitney test were used for statistical analyses.

## Materials and methods
### Animal care and diets
All animal care and experimental procedures were conducted in accordance with the guidelines of the Brazilian College for Animal Experimentation and approved by the Institutional Animal Care and Use Committee (CEUA 5497-1/2020 and 6210-1/2023). Dual-reporter CX3CR1^(GFP/+)CCR2^(RFP/+) mice were generated by mating CX3CR1^GFP homozygous mice (JAX#005582) with CCR2^RFP homozygous mice (JAX#017586). Heterozygous CCR2^(RFP/+) and CX3CR1^(GFP/+) mice were generated by mating homozygous CCR2^RFP and CX3CR1^GFP homozygous mice, respectively, with C57BL/6J. Genotypes of these mice were identified by polymerase chain reaction (PCR). C57BL/6J mice were obtained from the Multidisciplinary Center for Biological Research (CEMIB) at the State University of Campinas (UNICAMP). Mice were fed on standard chow diet (Nuvilab; 3.76 kcal/g; 12.6% energy from protein, 77.7% energy from carbohydrate, and 9.58% energy from fat) or HFD (5.28 kcal/g; 12.88% energy from protein, 27.1% energy from carbohydrate, and 60% energy from fat) according to the experimental protocols. Food and water were available ad libitum throughout the experimental periods, except for the protocols that required fasting. The room temperature was controlled (22–24°C), and a light–dark cycle was maintained on a 12-hr on–off cycle.

### Flow cytometry
For the separation of CX3CR1^(GFP+) and CCR2^(RFP+) cells from the WAT of CX3CR1^(GFP/+)CCR2^(RFP/+) mice we collected the retroperitoneal fat depot of one animal fed on a HFD for 4 weeks. It was minced and digested with type VIII collagenase (0.5 mg/ml, Sigma-Aldrich) in phosphate-buffered saline (PBS) for 20 min at 37°C with shaking. After digestion, the suspension was filtered using a 100-µm cell filter. For isolation of the same cells from the hypothalamus, samples of five CX3CR1^(GFP/+)CCR2^(RFP/+) mice fed on a HFD for 4 weeks were pooled together and gently pressed through a cell strainer (100 µm). The cell solution was subjected to a Percoll gradient (70/40%) for monocyte purification. Samples were acquired on a BDFacs Symphony instrument (BD Biosciences, USA) and then analyzed using FlowJo software.

### Cell sorting
For cell sorting of CX3CR1^(GFP+) and CCR2^(RFP+) cells from the hypothalamus we employed CX3CR1^(GFP/+) heterozygous mice fed on chow diet and dual-reporter CX3CR1^(GFP/+)CCR2^(RFP/+) mice fed on HFD for 4 weeks. Harvested hypothalami of 20–30 male or 20–30 female mice were pooled together for each sample and gently pressed through a cell strainer (100 µm). The cell solution was subjected to a Percoll gradient (70/40%) for monocyte purification. The sorting was conducted on a BDFacs Melody instrument (BD Biosciences, USA).

### RNA-sequencing and analysis
Cell-sorted CX3CR1^(GFP+) and CCR2^(RFP+) cell samples from hypothalamus were lysed for RNA extraction using the RNAqueous Micro kit (Invitrogen). RNA integrity was analyzed on a Bioanalyzer RNA Pico 6000 chip at the Core Facility for Scientific Research – University of São Paulo (CEFAP-USP). Low input RNA-Seq library preparation (Takara SMART-Seq v4) and sequencing by Illumina NovaSeq S2 PE150 Sequencing Lane (40M read pairs/sample avg) were performed by Maryland Genomics (Institute for Genome Sciences – IGS, University of Maryland School of Medicine – Baltimore, USA). Illumina sequencing adapters and low-quality reads were removed with Trimmomatic. Trimmed reads were aligned to the mouse reference genome (GRCm39) by STAR. Aligned reads were mapped to features using HTSeq, and differential expression analyses were performed using the DESeq2 package. Genes

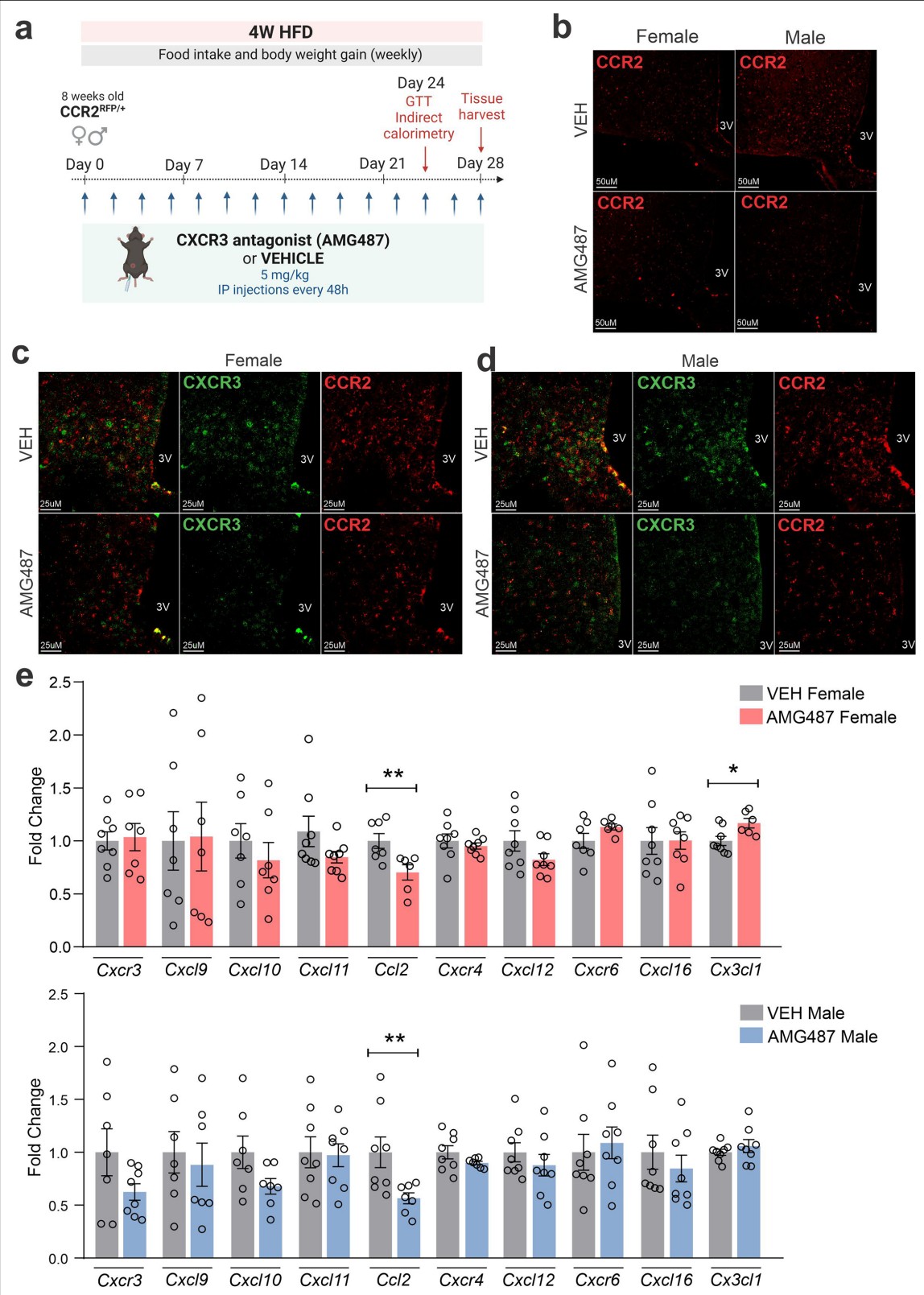

**Figure 10.** AMG487 treatment attenuates CCR2- and CXCR3-positive cell chemotaxis toward the hypothalamus of high-fat diet (HFD)-fed mice. (**a**) Schematic representation of the experimental protocol for CXCR3 systemic blockage. (**b**) Coronal brain sections from 4 weeks HFD-fed CCR2<sup>RFP/+</sup> mice showing the CCR2-positive cells distribution in the hypothalamic parenchyma upon AMG487 treatment. 3V = third ventricle, scale bars = 50 μm. (**c**) Coronal brain sections from 4 weeks HFD-fed female CCR2<sup>RFP/+</sup> mice immunostained for CXCR3 upon AMG487 treatment. (**d**) Coronal brain

*Figure 10 continued on next page*

*Figure 10 continued*

sections from 4 weeks HFD-fed male CCR2^RFP/+ mice immunostained for CXCR3 upon AMG487 treatment. 3V = third ventricle, scale bars = 25 µm. (**e**) Hypothalamic mRNA levels of several chemokine receptors and chemokines in HFD-fed female (light gray and pink bars) and male (light gray and blue bars) CCR2^RFP/+ mice upon AMG487 treatment. For qualitative confocal image analysis, we have used 3 samples per group. For RT-qPCR of the hypothalamus, we have used 7–8 samples per group. Two-tailed Mann–Whitney tests were used for statistical analyses. *p < 0.05 and **p < 0.01 in comparison with respective VEH-treated groups.

having less than 3 CPM were excluded before statistical analysis, and differentially expressed genes (DEGs) were selected using as cutoffs the adjusted p-value <0.05. Heatmaps were performed using pheatmap and a list of DEGs was passed to enrichR and cluster Profiler for enrichment analyses.

## CXCL10 immunoneutralization

For central neutralization of CXCL10, 8-week CCR2^RFP/+ heterozygous male and female mice underwent a stereotaxic surgery for ICV injections of anti-CXCL10 Monoclonal Antibody (2 µl, Cat# MA5-23774, Thermo Fisher). The control groups were ICV injected with Mouse IgG2a Isotype Control (2 µl, Cat#02-6200, Thermo Fisher). Two distinct ICV injections were performed on Days 0 and 14, respectively, of the experimental protocol. For that, mice were anesthetized with ketamine (100 mg/kg) and xylazine (10 mg/kg) and submitted to stereotaxic surgery (Ultra Precise–model 963, Kopf). ICV coordinates were [antero-posterior/lateral/depth to bregma]: −0.46/−1.0/−2.3 mm. Immediately after the first surgery, at Day 0, mice began to be fed on HFD for 4 weeks. From Day 0 to 28, food intake and body weight were evaluated weekly.

## CXCR3 antagonism

For systemic blockage of CXCR3, 8-week CCR2^RFP/+ heterozygous male and female mice underwent a treatment with AMG487 (Tocris Bioscience, Bristol, UK), an active and selective CXC chemokine receptor 3 (CXCR3) antagonist. The in vivo formulation of AMG487 was prepared in 20% hydroxypropyl-β-cyclodextrin (Sigma, St. Louis, MO). A 50% hydroxypropyl-β-cyclodextrin (Sigma, St. Louis, MO) solution was prepared and AMG487 was added to this solution, it was incubated in a sonicating water bath for 2 hr with occasional vortexing. Next, distilled water was added to give the appropriate final concentration of AMG487 in 20% of hydroxypropyl-β-cyclodextrin. This solution at 20% served as the vehicle. Mice were treated with AMG487 or vehicle (VEH group) intraperitoneally at 5 mg/kg every 48 hr throughout 4 weeks. During this period, mice were fed on HFD, and food intake and body weight were evaluated weekly.

## Ovariectomy procedure and estradiol replacement

Female C57BL/6J mice were anesthetized with ketamine (100 mg/kg) and xylazine (10 mg/kg). The ventral abdominal area was shaved and sterilized using an iodine solution. A small midline incision was made, and the ovaries were carefully located and excised bilaterally. The incision was then sutured using a suture thread. The Sham group, which underwent the same procedure except for ovary excision, was used as the control. Half of the ovariectomized mice also received estradiol replacement therapy. 17β-Estradiol pellets (0.05 mg/pellet, 60-day sustained release; Innovative Research of America, Inc, USA) were implanted subcutaneously beneath the dorsal surface of the neck. Tramadol hydrochloride (5 mg/kg, intraperitoneally) was administered immediately post-surgery, as well as 24 and 36 hr after surgery to manage pain. Mice were monitored during a 7-day recovery period. From Day 7 to 35, all groups were fed a HFD, and food intake and body weight were evaluated weekly. On the 25th day of the protocol, mice were fasted for 6 hr. Before euthanasia, we measured fasting glycemia. Afterward, we harvested the hypothalamus and the retroperitoneal WAT. The WAT was weighed for adiposity measurement, and the hypothalamus was used for qPCR analysis of chemokines, chemokine receptors, neuropeptides, and some inflammatory markers.

## Glucose tolerance test

On the 24th day of CXCR3 blockage and CXCL10 neutralization experimental protocols, mice were fasted for 6 hr, and blood glucose was measured via tail bleed at baseline and 15, 30, 60, 90, and 120 min after an intraperitoneal injection of glucose (2.0 g/kg).

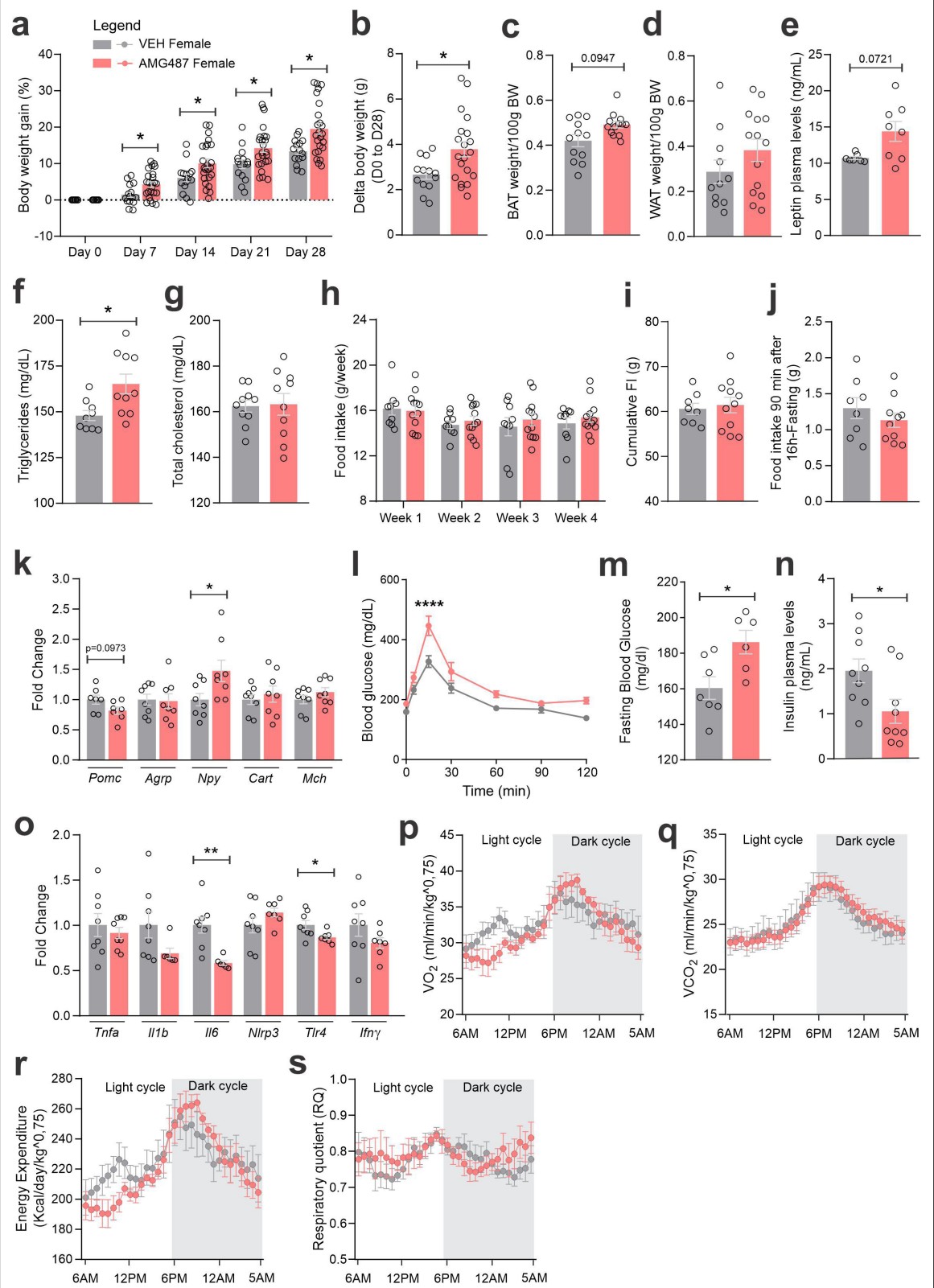

**Figure 11.** CXCR3 systemic blockage in high-fat diet (HFD)-fed female mice. (**a**) Percentual of body weight gain from Day 0 to 28 of the experimental protocol. (**b**) Delta body weight during the experimental period. (**c**) Brown adipose tissue weight and (**d**) white adipose tissue (retroperitoneal depot) weight at Day 28. (**e**) Leptin, (**f**) triglycerides, and (**g**) total cholesterol plasma levels at Day 28. (**h**) Weekly food intake measurement during experimental period. (**i**) Cumulative food intake during the experimental period. (**j**) 90-min food intake measurement after 16-hr fasting. (**k**) Hypothalamic mRNA levels

*Figure 11 continued on next page*

*Figure 11 continued*

of neuropeptides involved in food intake control. (**l**) Intraperitoneal glucose tolerance test at Day 24. (**m**) 6-hr fasting blood glucose levels. (**n**) Insulin plasma levels at Day 28. (**o**) Hypothalamic mRNA levels of inflammatory genes. (**p**) $O_2$ consumption; (**q**) $CO_2$ production; (**r**) energy expenditure; (**s**) respiratory quotient at Day 24. Data were expressed as mean ± SEM of 14–16 mice/group (in four independent experiments). To perform quantitative reverse transcription-polymerase chain reaction (qRT-PCR) we have used 8 mice/group. To perform biochemical analysis in plasma we have used 8–10 mice/group. To perform ipGTT we have used 5 mice/group. To perform indirect calorimetry, we have used 4–5 mice/group. Two-way ANOVA followed by Sidak's post hoc test and Mann–Whitney test were used for statistical analyses. $*p < 0.05$, $**p < 0.01$, $****p < 0.0001$ in comparison with VEH-treated group.

The online version of this article includes the following figure supplement(s) for figure 11:

**Figure supplement 1.** CXCR3 systemic blockage in high-fat diet (HFD)-fed female mice.

## Indirect calorimetry and locomotor activity

The oxygen consumption ($VO_2$), carbon dioxide production ($VCO_2$), energy expenditure, and respiratory quotient were measured using an indirect open-circuit calorimeter (Oxylet M3 system; PanLab/Harvard Apparatus, MA, USA). Spontaneous locomotor activity was measured using a Panlab Infrared (IR) Actimeter, which consists of a two-dimensional (*X* and *Y* axes) square frame, a frame support, and a control unit. Each frame is equipped with 16 × 16 infrared beams for optimal subject detection (PanLab/Harvard Apparatus, MA, USA). For each mouse, we calculated the mean of total movements per hour over 24 hr. Mice were allowed to adapt for 12 hr before data were recorded for 24 hr (light and dark cycles).

## Immunofluorescence

On the day 28th of CXCR3 blockage and CXCL10 neutralization experimental protocols, male and female mice were perfused with 0.9% saline followed by 4% formaldehyde by cardiac cannulation. Brains were extracted and incubated in 4% formaldehyde overnight at 4°C for extended fixation. The brains were then incubated in 30% sucrose at 4°C for 48 hr. A series of 20-μm-thick frozen sections (four series equally) were prepared using a cryostat and stored in an anti-freezing solution. For the free-floating immunostaining, slices were washed with 0.1 M PBS (three times, 5 min each) and blocked with 0.2% Triton X-100 and 5% donkey serum in 0.1 M PBS for 2 hr at room temperature. Slices were incubated overnight at 4°C with Anti-Cxcr3 (1:200, Cat# NB100-56404, Novus Biologicals) or Anti-Sialoadhesin/CD169 (1:200, ab18619, Abcam) in a blocking solution. After washing with 0.1 M PBS (three times, 5 min each), sections were incubated with fluorophore-labeled secondary antibody (donkey anti-rabbit Alexa Fluor 405, 1:500, Cat# A48258, Invitrogen or goat anti-mouse Alexa Fluor 405, 1:500, Cat# A31553, Invitrogen) in a blocking solution for 2 hr at room temperature. After washing again with 0.1 M PBS (three times, 5 min each), brain slices were mounted onto slides with ProLong Diamond antifade mountant (Cat# P36930, Thermo Fisher). Sections were visualized with a Zeiss LSM780, confocal microscope (Carl Zeiss AG, Germany) at the National Institute of Photonics Applied to Cell Biology (INFABIC) at the University of Campinas.

## Quantitative reverse transcription-polymerase chain reaction

Total RNA was extracted using a TRIzol reagent (Thermo Fisher Scientific) and synthesized cDNA with a High-Capacity cDNA Reverse Transcription Kit (HighCapacity cDNA Reverse Transcription Kit, Life Technologies). Real-time PCR reactions were run using the TaqMan system (Thermo Fisher Scientific). Primers used were *Cxcr3* (Mm99999054_s1); *Cxcl9* (Mm00434946_m1), *Cxcl10* (Mm00445235_m1); *Cxcl11* (Mm00444662_m1); *Ccl2* (Mm00441242_m1); *Cxcr4* (Mm01996749_s1); *Cxcl12* (Mm00445553_m1), *Cxcr6* (Mm02620517_s1), *Cxcl16* (Mm00469712_m1), *Cx3cl1* (Mm00436454_m1), *Pomc* (Mm00435874_m1), *Agrp* (Mm00475829_g1); *Npy* (Mm00445771_m1); *Cartpt* (Mm04210469_m1), *Pmch* (Mm01242886_g1), *Tnfa* (Mm00443258_m1), *Il1b* (Mm00434228_m1), *Il6* (Mm00446190_m1), *Nlrp3* (Mm00840904_m1), *Tlr4* (Mm00445273_m1), *Ifng* (Mm01168134_m1), *Eif2a* (Mm01289723_m1), *Atf6* (Mm01295319_m1), *Ddit3* (Mm01135937_g1), *Immp2l* (Mm00474144_m1), *Mfn1* (Mm00612599_m1), *Opa1* (Mm01349707_g1), *Htra2* (Mm00444846_g1), *Ppargc1a* (Mm01208835_m1), *FasN* (Mm00662319_m1), *Scd1* (Mm00772290_m1), *Scd2* (Mm01208542_m1), *Pck1* (Mm01247058_m1), *G6pc3* (Mm00616234_m1), *G6pc* (Mm04207416_m1), *Pparg* (Mm00440940_m1), *Prdm16* (Mm00712556_m1), *Ucp1* (Mm01244861_m1). *Gapdh* (Mm99999915_g1) was employed as

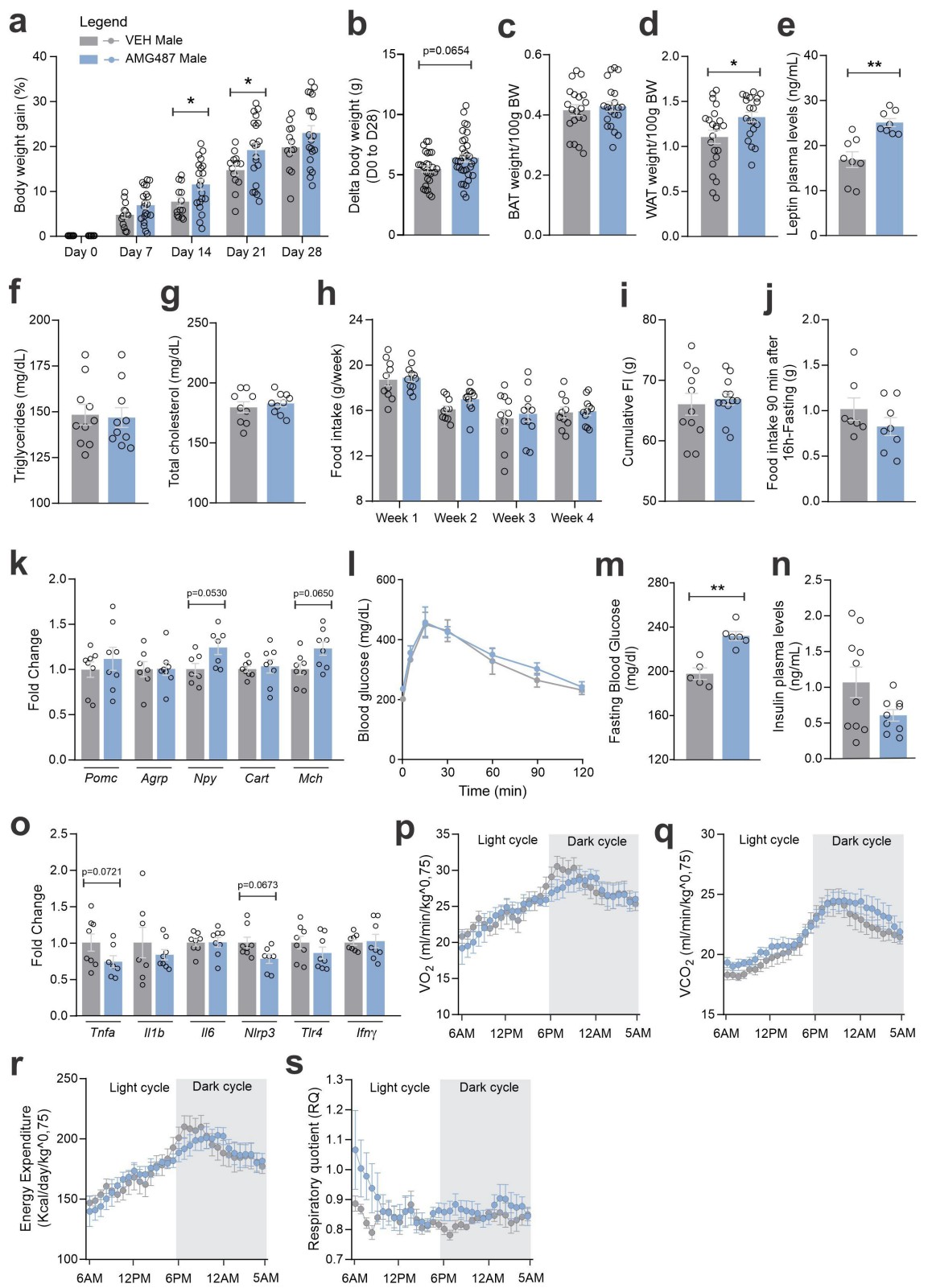

**Figure 12.** CXCR3 systemic blockage in high-fat diet (HFD)-fed male mice. (**a**) Percentual of body weight gain from Day 0 to 28 of the experimental protocol. (**b**) Delta body weight during the experimental period. (**c**) Brown adipose tissue weight and (**d**) white adipose tissue (retroperitoneal depot) weight at Day 28. (**e**) Leptin, (**f**) triglycerides, and (**g**) total cholesterol plasma levels at Day 28. (**h**) Weekly food intake measurement during the experimental period. (**i**) Cumulative food intake during the experimental period. (**j**) 90-min food intake measurement after 16 hr of fasting.

*Figure 12 continued on next page*

*Figure 12 continued*

(**k**) Hypothalamic mRNA levels of neuropeptides involved in food intake control. (**l**) Intraperitoneal glucose tolerance test on Day 24. (**m**) 6-hr fasting blood glucose levels. (**n**) Insulin plasma levels at Day 28. (**o**) Hypothalamic mRNA levels of inflammatory genes. (**p**) $O_2$ consumption; (**q**) $CO_2$ production; (**r**) energy expenditure; (**s**) respiratory quotient at Day 24. Data were expressed as mean ± SEM of 14–16 mice/group (in four independent experiments). To perform quantitative reverse transcription-polymerase chain reaction (qRT-PCR) we have used 8 mice/group. To perform biochemical analysis in plasma we have used 8–10 mice/group. To perform ipGTT we have used 5 mice/group. To perform indirect calorimetry, we have used 4–5 mice/group. Two-way ANOVA followed by Sidak's post hoc test and Mann–Whitney test were used for statistical analyses. *$p < 0.05$, **$p < 0.01$ in comparison with VEH-treated group.

The online version of this article includes the following figure supplement(s) for figure 12:

**Figure supplement 1.** CXCR3 systemic blockage in high-fat diet (HFD)-fed male mice.

the reference gene for all tissues, except for WAT, where *Actb* (Mm02619580_g1) was employed as the reference gene. Gene expression was obtained using QuantStudio 6 (Thermo Fisher Scientific).

## Hormonal and biochemical determinations

Serum insulin, leptin, and estradiol were measured by enzyme-linked immunosorbent assay (ELISA) kits (#EZRMI-13K and #EZML-82K; Millipore; E-EL-0150; Elabscience). Serum triglyceride levels and total cholesterol were measured using a commercial colorimetric assay kit (LaborLab, Guarulhos – SP, Brazil) following the manufacturer's instructions.

## Statistical analysis

Data are presented as means ± standard error of the mean. The statistical analyses were carried out using a non-parametric Mann–Whitney test and one- or two-way analysis of variance when appropriate. Post hoc comparisons were performed using Sidak's test. Statistical significances were analyzed using Prism 8.0 software (GraphPad Software, La Jolla, CA). A p-value ≤0.05 was considered statistically significant. In the experiments aimed at measuring energy expenditure, data were always corrected for body mass.

## Acknowledgements

We are grateful to Erika Roman, Joseane Morari, Marcio Cruz, and Gerson Ferraz for laboratory management. This research was funded by The Sao Paulo Research Foundation (FAPESP): 2013/07607-8; 2017/22511-8; and 2021/00443-6.

## Additional information

### Funding

| Funder | Grant reference number | Author |
|---|---|---|
| Fundação de Amparo à Pesquisa do Estado de São Paulo | 2013 07607-8 | Licio A Velloso |
| Fundação de Amparo à Pesquisa do Estado de São Paulo | 2017/22511-8 | Natalia Mendes |
| Fundação de Amparo à Pesquisa do Estado de São Paulo | 2021/00443-6 | Natalia Mendes |

The funders had no role in study design, data collection, and interpretation, or the decision to submit the work for publication.

### Author contributions

Natalia Mendes, Conceptualization, Data curation, Formal analysis, Investigation, Methodology, Writing – original draft, Writing – review and editing; Ariane Zanesco, Formal analysis, Investigation,

Methodology, Writing – review and editing; Cristhiane Aguiar, Gabriela F Rodrigues-Luiz, Dayana Silva, Jonathan Campos, Investigation, Methodology, Writing – review and editing; Niels Olsen Saraiva Camara, Resources, Writing – review and editing; Pedro Moraes-Vieira, Resources, Supervision, Writing – review and editing; Eliana Araujo, Conceptualization, Formal analysis, Supervision, Writing – original draft, Project administration, Writing – review and editing; Licio A Velloso, Conceptualization, Resources, Data curation, Formal analysis, Supervision, Funding acquisition, Validation, Visualization, Writing – original draft, Project administration, Writing – review and editing

**Author ORCIDs**
Pedro Moraes-Vieira ⓘ https://orcid.org/0000-0002-8263-786X
Licio A Velloso ⓘ https://orcid.org/0000-0002-4806-7218

**Ethics**
All animal care and experimental procedures were conducted in accordance with the guidelines of the Brazilian College for Animal Experimentation and approved by the Institutional Animal Care and Use Committee (CEUA 5497-1/2020 and 6210-1/2023).

Reviewer #1 (Public review): https://doi.org/10.7554/eLife.95044.3.sa1
Reviewer #2 (Public review): https://doi.org/10.7554/eLife.95044.3.sa2
Author response https://doi.org/10.7554/eLife.95044.3.sa3

## Additional files

**Supplementary files**
• MDAR checklist

**Data availability**
The RNA-sequencing data for this study were submitted to the NCBI Sequence Read Archive (SRA) under BioProject accession number PRJNA1155598.

The following dataset was generated:

| Author(s) | Year | Dataset title | Dataset URL | Database and Identifier |
| --- | --- | --- | --- | --- |
| Mendes N | 2024 | Bulk sequence of hypothalamus | https://www.ncbi.nlm.nih.gov/bioproject/?term=PRJNA1155598 | NCBI BioProject, PRJNA1155598 |

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
