## [Editor Report · eLife Assessment]

This work is of **fundamental** significance and has an **exceptional** level of evidence for a new population that protects against obesity-induced hypothalamic inflammation. This topic will attract attention from a broad base of readers, from hypothalamic neuroscientists to immunologists with an interest in metabolism.

---

## [Referee Report · Reviewer #1 (Public review)]

Summary:

The present work from Velloso and collaborators investigated the transcription profiles of resident and recruited hypothalamic microglia. They found sex-dependent differences between males and females and identified the protective role of chemokine receptor CXCR3 against diet-induced obesity.

Strengths:

(1) Novelty

(2) Relevance, since this work provides evidence about a subset of recruited microglia that has a protective effect against DIO. This provides a new concept in hypothalamic inflammation and obesity.

Comments on revised version:

All my comments have been addressed.

---

## [Referee Report · Reviewer #2 (Public review)]

Summary:

This study by Mendes et al provides novel key insights in the role of chemotaxis and immune cell recruitment into the hypothalamus in the development of diet-induced obesity. Specifically, the authors first revealed that although transcriptional changes in hypothalamic resident microglia following exposure to high-fat feeding are minor, there are compelling transcriptomic differences between resident microglia and microglia recruited to the hypothalamus, and these are sexually dimorphic. Using independent loss-of-function studies, the authors also demonstrate an important role of CXCR3 and hypothalamic CXCL10 in the hypothalamic recruitment of CCR2+ positive cells on metabolism following exposure to high-fat diet-feeding in mice. This manuscript puts forth conceptually novel evidence that inhibition of chemotaxis-mediated immune cell recruitment accelerates body weight gain in high-fat diet-feeding, suggesting that a subset of microglia which express CXCR3 may confer protective, anti-obesogenic effects.

Strengths:

The work is exciting and relevant given the prevalence of obesity and the consequences of inflammation in the brain on perturbations of energy metabolism and ensuant metabolic diseases. Hypothalamic inflammation is associated with disrupted energy balance, and activated microglia within the hypothalamus resulting from excessive caloric intake and saturated fatty acids are often thought to be mediators of impairment of hypothalamic regulation of metabolism. The present work reports a novel notion in which immune cells recruited into the hypothalamus which express chemokine receptor CXCR3 may have a protective role against diet-induced obesity. In vivo studies reported herein demonstrate that inhibition of CXCR3 exacerbates high-fat diet-induced body weight gain, increases circulating triglycerides and fasting glucose levels, worsens glucose tolerance, and increases the expression of orexigenic neuropeptides, at least in female mice.

This work provides a highly interesting and needed overview of preclinical and clinical brain inflammation, which is relevant to readers with an interest in metabolism and immunometabolism in the context of obesity.

Using flow cytometry, cell sorting, and transcriptomics including RNA-sequencing, the manuscript provides novel insights on transcriptional landscapes of resident and recruited microglia in the hypothalamus. Importantly, sex differences are investigated.

Overall, the manuscript is perceived to be highly interesting, relevant, and timely. The discussion is thoughtful, well-articulated, and a pleasure to read and felt to be of interest to a broad audience.

Weaknesses:

There were no major weaknesses perceived. Some comments for potential textual additions to the results/discussion are provided below.

Could the authors comment on the choice of peripheral administration of CXCR3 antagonist as opposed to central (e.g. icv) administration? Indeed, systemic inhibition of CXCR3 produced significant alterations in body weight gain and glucose tolerance in female mice given high-fat diet and reduced CCR2 and CXCR3 immunostaining in the hypothalamus. Could changes to peripheral (e.g. WAT, liver) immune responses to the diet underlie the metabolic changes observed?

Besides hypothalamic mRNA levels of chemokines and chemokine receptors, does systemic CXCR3 antagonism affect other aspects linked to diet-induced impairments of hypothalamic regulation of energy homeostasis, like inflammation, ER stress and/or mitochondrial dynamics/function? It would be interesting to reveal the consequence of reduced CCR2+ microglial migration to the hypothalamus with chronic high-fat diet exposure.

---

## [Author Response]

The following is the authors’ response to the original reviews.

**Public Reviews:**

**Reviewer #1 (Public Review):**
Summary:The present work from Velloso and collaborators investigated the transcription profiles of resident and recruited hypothalamic microglia. They found sex-dependent differences between males and females and identified the protective role of chemokine receptor CXCR3 against diet-induced obesity.Strengths:(1) Novelty;(2) Relevance, since this work provides evidence about a subset of recruited microglia that has a protective effect against DIO. This provides a new concept in hypothalamic inflammation and obesity.Weaknesses:(1) Lack of mechanistic insight into the sex-dependent effects;(2) Analysis of indirect calorimetry data requires more depth;(3) A deeper analysis of hypothalamic inflammation and ER stress pathways would strengthen the manuscript.
**Reviewer #2 (Public Review):**
Summary:This study by Mendes et al provides novel key insights into the role of chemotaxis and immune cell recruitment into the hypothalamus in the development of diet-induced obesity. Specifically, the authors reveal that although transcriptional changes in hypothalamic resident microglia following exposure to high-fat feeding are minor, there are compelling transcriptomic differences between resident microglia and microglia recruited to the hypothalamus, and these are sexually dimorphic. Using independent loss-of-function studies, the authors also demonstrate an important role of CXCR3 and hypothalamic CXCL10 in the hypothalamic recruitment of CCR2+ positive cells on metabolism following exposure to high-fat diet-feeding in mice. This manuscript puts forth conceptually novel evidence that inhibition of chemotaxis-mediated immune cell recruitment accelerates body weight gain in high-fat diet-feeding, suggesting that a subset of microglia that express CXCR3 may confer protective, anti-obesogenic effects.Strengths:The work is exciting and relevant given the prevalence of obesity and the consequences of inflammation in the brain on perturbations of energy metabolism and ensuant metabolic diseases. Hypothalamic inflammation is associated with disrupted energy balance, and activated microglia within the hypothalamus resulting from excessive caloric intake and saturated fatty acids are often thought to be mediators of impairment of hypothalamic regulation of metabolism. The present work reports a novel notion in which immune cells recruited into the hypothalamus that express chemokine receptor CXCR3 may have a protective role against diet-induced obesity. In vivo studies reported herein demonstrate that inhibition of CXCR3 exacerbates high-fat diet-induced body weight gain, increases circulating triglycerides and fasting glucose levels, worsens glucose tolerance, and increases the expression of orexigenic neuropeptides, at least in female mice.This work provides a highly interesting and needed overview of preclinical and clinical brain inflammation, which is relevant to readers with an interest in metabolism and immunometabolism in the context of obesity.Using flow cytometry, cell sorting, and transcriptomics including RNA-sequencing, the manuscript provides novel insights into transcriptional landscapes of resident and recruited microglia in the hypothalamus. Importantly, sex differences are investigated.Overall, the manuscript is perceived to be highly interesting, relevant, and timely. The discussion is thoughtful, well-articulated, and a pleasure to read and felt to be of interest to a broad audience.Weaknesses:There were no major weaknesses perceived. Some comments for potential textual additions to the results/discussion are listed in recommendations to authors.

Comments from the authors regarding the evaluation of the article: We publicly express our gratitude for the work of both Reviewers. The comments were timely and constructive and guided us toward preparing a new version of the article which contains novel data that strengthened the overall quality of the study.

**Recommendations for the authors:**

**Reviewer #1 (Recommendations For The Authors):**
Experiments with ovariectomized female mice with (and without) estrogen replacement would help to address the physiological basis of the observed sexdependent effects.

We performed an experiment with female C57BL/6J Unib, subdivided into Sham, OVX, and OVX+EST groups, which were exposed to HFD for 4 weeks. We monitored the weekly evolution of body weight and food intake. At the end of the protocol, the animals fasted for 4 hours. Then, we measured fasting blood glucose and estradiol; and extracted tissues (hypothalamus andWAT). In the hypothalamus samples, we evaluated, by RT-qPCR, the expression of chemokines, chemokine receptors, and some pro-inflammatory cytokines and neuropeptides. We evaluated the body mass relative WAT weight. The new results are presented in Supplementary Figure 1.

Indirect calorimetric analysis of energy expenditure will benefit from ANCOVA analysis using body weight as a covariate. Moreover, locomotor activity should be also controlled.

All statistical analysis regarding energy expenditure is corrected by body mass, thus, there is no need for ANCOVA, we clarified this in the text. The determination of locomotor activity is now included in Supplementary Figures 2 and 3.

A deeper analysis of hypothalamic inflammation and ER stress pathways would strengthen the manuscript.

We performed new experiments to determine the expression of hypothalamic inflammation and ER stress pathaways. This is shown in Suppl. Fig. 2 and 3.

Mechanistic inhibition of CXCR3 was performed by CXCL10 immunoneutralization and CXCR3 antagonism. Those approaches are correct and well-performed, however considering the experience of the group in hypothalamic studies, I miss a virogenetic-based knockdown. Do the authors have any data on that?

This is indeed a great point. Unfortunately, we did not succeed in obtaining mice Cre lineages that would be needed for the proposed experiments. We included this as a weakness of the study.

**Reviewer #2 (Recommendations For The Authors):**
There are a few typographical errors for correction:- Page 4, line 157: CCL10 to CXCL10.- Page 6, line 226: makers to markers.- Page 7, lines 283 and 287, Figure 6C: INF to IFN.

All errors were corrected, as recommended.

Parts of the manuscript may be difficult for readers without knowledge of transcriptomics to interpret; thus, further description of several of the figures (e.g. Figure 3 and 4) may be helpful.

We expanded the text in Results to clarify this issue.

Could the authors comment on the choice of peripheral administration of CXCR3 antagonist as opposed to central (e.g. icv) administration? Indeed, systemic inhibition of CXCR3 produced significant alterations in body weight gain and glucose tolerance in female mice given high-fat diets and reduced CCR2 and CXCR3 immunostaining in the hypothalamus. Could changes to peripheral (e.g. WAT, liver) immune responses to the diet underlie the metabolic changes observed?

CXCR3+ cells are present in very small numbers in the hypothalamus under basal conditions. In HFD, these are recruited from the periphery to the CNS, so, we believe ICV treatment with AMG487 would not reduce recruitment to the hypothalamic parenchyma. With the same animals in which we performed the locomotor activity, we performed RT-qPCR of WAT and liver and analyzed the expression of genes involved in lipid and glucose metabolism. This is now in Supplementary Figures 2 and 3. We included a comment in the text to explain our rationale for this approach.

Besides hypothalamic mRNA levels of chemokines and chemokine receptors, does systemic CXCR3 antagonism affect other aspects linked to diet-induced impairments of hypothalamic regulation of energy homeostasis, like inflammation, ER stress and/or mitochondrial dynamics/function? It would be interesting to reveal the consequence of reduced CCR2+ microglial migration to the hypothalamus with chronic high-fat diet exposure.

We performed new experiments shown in Supplementary Figures 2 and 3 to deal with these important questions. In the hypothalamus of females there were no changes in the expression of transcripts encoding proteins involved in endoplasmic reticulum homeostasis and mitochondrial turnover, whereas in males there was a reduction of Ddit3 and Mfn1. Moreover, in females the inhibition of CXCR3 promoted no changes in the liver expression of lipidogenic and gluconeogenic genes, and no changes in the white adipose tissue expression of lipidogenic genes. In the liver of males, there was a reduction in the expression of Fasn and an increase in the expression of G6pc3. As for the females, in males, there were no changes in the white adipose tissue expression of lipidogenic genes.